# Sequence and structural conservation reveal fingerprint residues in TRP channels

Deny Cabezas-Bratesco[1†], Francisco A Mcgee[2†], Charlotte K Colenso[1,3], Kattina Zavala[4], Daniele Granata[2], Vincenzo Carnevale[2], Juan C Opazo[4,5,6]*, Sebastian E Brauchi[1,6,7]*

[1]Instituto de Fisiologia, Facultad de Medicina, Universidad Austral de Chile, Valdivia, Chile; [2]Institute for Computational Molecular Science and Department of Biology, Temple University, Philadelphia, United States; [3]School of Cellular and Molecular Medicine, University of Bristol, Bristol, United Kingdom; [4]Instituto de Ciencias Ambientales y Evolutivas, Facultad de Ciencias, Universidad Austral de Chile, Valdivia, Chile; [5]Integrative Biology Group, Universidad Austral de Chile, Valdivia, Chile; [6]Millennium Nucleus of Ion Channel-associated Diseases (MiNICAD), Valdivia, Chile; [7]Janelia Research Campus, Howard Hughes Medical Institute, Ashburn, United States

*For correspondence:
jopazo@gmail.com (JCO);
sbrauchi@uach.cl (SEB)

[†]These authors contributed equally to this work

Competing interest: The authors declare that no competing interests exist.

**Abstract** Transient receptor potential (TRP) proteins are a large family of cation-selective channels, surpassed in variety only by voltage-gated potassium channels. Detailed molecular mechanisms governing how membrane voltage, ligand binding, or temperature can induce conformational changes promoting the open state in TRP channels are still a matter of debate. Aiming to unveil distinctive structural features common to the transmembrane domains within the TRP family, we performed phylogenetic reconstruction, sequence statistics, and structural analysis over a large set of TRP channel genes. Here, we report an exceptionally conserved set of residues. This fingerprint is composed of twelve residues localized at equivalent three-dimensional positions in TRP channels from the different subtypes. Moreover, these amino acids are arranged in three groups, connected by a set of aromatics located at the core of the transmembrane structure. We hypothesize that differences in the connectivity between these different groups of residues harbor the apparent differences in coupling strategies used by TRP subgroups.

## Editor's evaluation

This study by Deny Cabezas-Bratesco and collaborators draws from multiple bioinformatics approaches, as well as from published structural and functional data, to uncover a set of highly conserved amino acid sequence features in group I TRP ion channels. These identified features provide insight into the evolution and mechanisms of function of this diverse and important family of ion channel proteins.

## Introduction

Transient receptor potential (TRP) proteins constitute a large family of cation-selective ion channels involved in a number of physiological functions (*Clapham, 2003*; *Nilius et al., 2007*). Changes in TRP channel function and expression are associated with a variety of metabolic, respiratory, cardiovascular, and neurological diseases (*Nilius et al., 2007*; *Nelson et al., 2011*; *Wang et al., 2020*). Moreover,

the abnormal expression of TRP channels has been related to cancer development and progression (*Shapovalov et al., 2016*; *Yang and Kim, 2020*). TRP channels are therefore an attractive target for pharmacological development, and the understanding of their inner workings is critical for such endeavors.

TRPs have been shown to be evolutionarily related to all voltage-gated cation channels (VGCCs), two-pore channels (TPCs or TPCNs), and CatSper channels (*Yu et al., 2005*). The TRP family is composed of two major groups (Groups I and II) and 10 subfamilies, or subtypes: TRPA1, TRPV, TRPVL, TRPC, TRPM, TRPS, TRPN, TRPY/TRPF PKD2s, and TRPML (*Ramsey et al., 2006*). While Group I (GI) gathers members of TRPC, TRPM, TRPS, TRPV, TRPVL, TRPA, and TRPN subfamilies; Group II (GII) is composed of members of the TRPML, PKD2, and TRPY/TRPF channels (*Venkatachalam and Montell, 2007*; *Himmel and Cox, 2020*; *Himmel et al., 2020*).

In general, TRP channels share poor cation selectivity and a loose sequence similarity (*Ramsey et al., 2006*). Cataloged as polymodal ion channels, they have the ability to integrate multiple stimuli (e.g., chemical, mechanical, electrical, and thermal) to promote channel opening. Such polymodality observed in TRPs has been explained in terms of allosteric interactions (*Brauchi et al., 2004*; *Latorre et al., 2009*). Different lines of research have shown that the different sensor modules couple to each other and to the channel pore, modulating permeation (*Hui et al., 2003*; *Castillo et al., 2018*; *Zubcevic et al., 2019*; *Zhao et al., 2020*; *Yang et al., 2018*; *Yang et al., 2020*).

Structural data revealed that TRP channels share the general architecture of voltage-gated ion channels (VGICs) (*Kasimova et al., 2016*; *Cheng, 2018*; *Cao, 2020*). They assemble as domain-swapped tetramers, with monomers containing six transmembrane segments (i.e., TM1–TM6) flanked by cytoplasmic N- and C-terminal domains. The transmembrane helices TM5 and TM6, together with the section between them, give shape to the conductive pore (*Ramsey et al., 2006*; *Liao et al., 2013*). The first four transmembrane helices (TM1–TM4) and the intracellular domains have been described as regulatory regions as they provide binding sites for agonists and cofactors (*Steinberg et al., 2014*; *Voolstra and Huber, 2014*; *Cao, 2020*). The TM1 and TM4 transmembrane region shows fundamental differences with known VGICs, where the absence of charged residues within the transmembrane region should be underscored (*Palovcak et al., 2015*; *Cao, 2020*). It has been shown that TRPMs and TRPVs are about tenfold less voltage-dependent compared to voltage-gated potassium channels. The modest voltage dependence observed in TRP channels is likely supported by residues located in the pore region (*Liu et al., 2009*; *Yang et al., 2020*). The TM1 and TM4 transmembrane region hosts binding pockets for ligands in all the different TRP channel subtypes, serving as a ligand-binding domain (LBD) in TRPs (*Steinberg et al., 2014*; *Huffer et al., 2020*). Nevertheless, this LBD has historically been referred to as the voltage-sensing-like domain (VSLD).

Regardless of the considerable variability in sequence similarity and physiological function, it is clear that TRP channels are closely related, especially those within Group I (*Kadowaki, 2015*; *Peng et al., 2015*; *Arias-Darraz et al., 2015*). The structural understanding of TRP conformational and functional behavior has been strengthened by technical advances allowing the recent release of a large set of high-resolution cryo-electron microscopy (cryo-EM) structures (*Cao, 2020*; *Samanta et al., 2019*). Led by the structure of TRPV1, structures have recently become available for at least one member of each subfamily, including crTRP1 from the green algae *Chlamydomonas reinhardtii* (*McGoldrick et al., 2019*; *Cao, 2020*). Together with advances in structural biology, intense research during recent years has been focused on understanding the molecular mechanisms supporting TRP activation and regulation (*Hofmann et al., 2017*; *Yang and Zheng, 2017*; *Yang et al., 2018*; *Singh et al., 2018a*; *Hilton et al., 2019*; *Zhang et al., 2019*; *Zubcevic et al., 2019*; *Zhao et al., 2020*; *Nadezhdin et al., 2021a*; *Nadezhdin et al., 2021a*). Although a consistent picture accounting for general principles governing TRP channels' mechanics is still missing, several structural features of great importance have been identified. Among these are the proximal N- and C-terminal domains flanking the transmembrane region, namely the pre-TM1 (also called pre-S1) and the TRP domain helix (TDh), respectively. These elements, seemingly related to the integration of molecular mechanics during activation, are present in all members of GI-TRPs and absent in GII-TRPs (*McGoldrick et al., 2019*; *Zhao et al., 2020*; *Nadezhdin et al., 2021b*).

Here, we studied TRP channel proteins from a large variety of organisms aiming to understand how they are evolutionarily related, and to find conserved structural elements. We obtained a well-resolved phylogeny, providing a snapshot of the TRP gene family duplicative history, and a phylogenetic

framework to understand the evolution of structural and functional attributes present in the TRP gene family. We also found 12 conserved and non-contiguous amino acids (W F Φ G Φ Φ Φ N L I A W) present in all TRP channels from GI-TRPs that we interpreted as a fingerprint.

Moreover, we discovered strong conservation unique to each TRP subtype. The amino acid conservation can be traced down to TRP channels from unicellular organisms, suggesting a robust architectural design. In addition, we identified a group of aromatic residues facing the core of the LBD (i.e., TM 1–4) in all TRP subtypes. In agreement with our phylogenetic reconstruction, this aromatic core (AC) can be observed in the unicellular crTRP1, it is absent in VGICs, and present in a rudimentary form in TPCs. Our structural analyses suggest that TRP channel specialization could be oriented around inter-subunit interactions between AC residues of one subunit with fingerprint residues at the selectivity filter of a neighboring subunit. Overall, our results suggest that TRP channel specialization has been built around the connectivity of heavily conserved distant residues that are located at critical sites within the structure.

## Results

### Phylogenetic diversification of the TRP gene family and the phylogenetic position of unicellular TRPs

Over millennia, the diversification of TRP channels produced a variety of lineages that we currently identify as subfamilies in an established nomenclature. The understanding of phylogenetic relationships among TRPs has been subject to an intense debate, and different phylogenetic hypotheses have been proposed (*Clapham et al., 2001*; *Clapham, 2003*; *Sidi et al., 2003*; *Yu and Catterall, 2004*; *Yu et al., 2005*; *Montell, 2005*; *Yu et al., 2005*; *Venkatachalam and Montell, 2007*; *Latorre et al., 2009*; *Nilius and Owsianik, 2011*; *Arias-Darraz et al., 2015*; *Ferreira et al., 2015*; *Peng et al., 2015*; *Eriksson et al., 2018*; *Kozma et al., 2018*; *Himmel et al., 2020*; *Himmel et al., 2020*; *Hsiao et al., 2021*). The diversity reported in phylogenetic topologies so far can be explained by the differences in taxonomic sampling, and to the fact that not all studies include all subfamilies and/or outgroups. To advance our understanding of such a process of diversification, we first performed a phylogenetic analysis, overcoming the caveats mentioned above by including representative members of each reported TRP subfamily in addition to outgroups (i.e., Kv and Nav channels).

Our gene phylogeny is well resolved and provides a snapshot of the TRP gene duplicative history (*Figure 1*; *Figure 1—figure supplement 1*). To validate this phylogeny, we repeated the phylogenetic analysis 10 times, observing that the evolutionary relationships among the main TRP lineages were consistent, with only negligible variation in the likelihood values. We not only recovered the monophyly (i.e., the single evolutionary origin of the TRP gene family) with strong support of 98%, but also the diversity of TRP channels into two main groups: (1) GI-TRPs, a clade containing TRPA1, TRPV, TRPVL, TRPC, TRPGamma, TRPN1, TRPY/TRPF, TRPM, and TRPS channels; and (2) GII-TRPs, a clade containing PKD2 and MCLN channels (*Figure 1*). We recovered four well-supported clades for GI-TRPs: (A) TRPV/TRPVL/TRPA1, (B) TRPC/TRPGamma/TRPN1, (C) TRPY/TRPF, and (D) TRPM/TRPS. Among these, we recovered the sister group relationship between the TRPV/TRPVL/TRPA1 and TRPC/TRPGamma/TRPN1 clades with strong support (*Figure 1*). Moreover, the TRPY/TRPF group was recovered sister to the latter (*Figure 1*). The TRPM/TRPS clade was recovered sister to all other members of GI-TRPs (*Figure 1*).

Next, we performed phylogenetic analyses to investigate the phylogenetic position of TRP channels from unicellular organisms, as they are most divergent. According to our analyses, a TRP sequence from *Coccomyxa subellipsoidea* was recovered sister to the TRPV/TRPVL/TRPA1 clade, while a TRP sequence from *C. reinhardtii* was recovered sister to the TRPM/TRPS clade (*Figure 1—figure supplements 2 and 3*). As reported before (*Arias-Darraz et al., 2015*), all other TRP sequences from unicellular organisms considered in this work are more evolutionarily related to the GII-TRP clade (*Figure 1—figure supplement 2*).

### Multiple sequence alignments identify a discrete set of highly conserved residues

We then focused our attention on the transmembrane regions and the immediate flanking segments that are common to all GI-TRPs. In particular, we analyzed the protein segment containing the

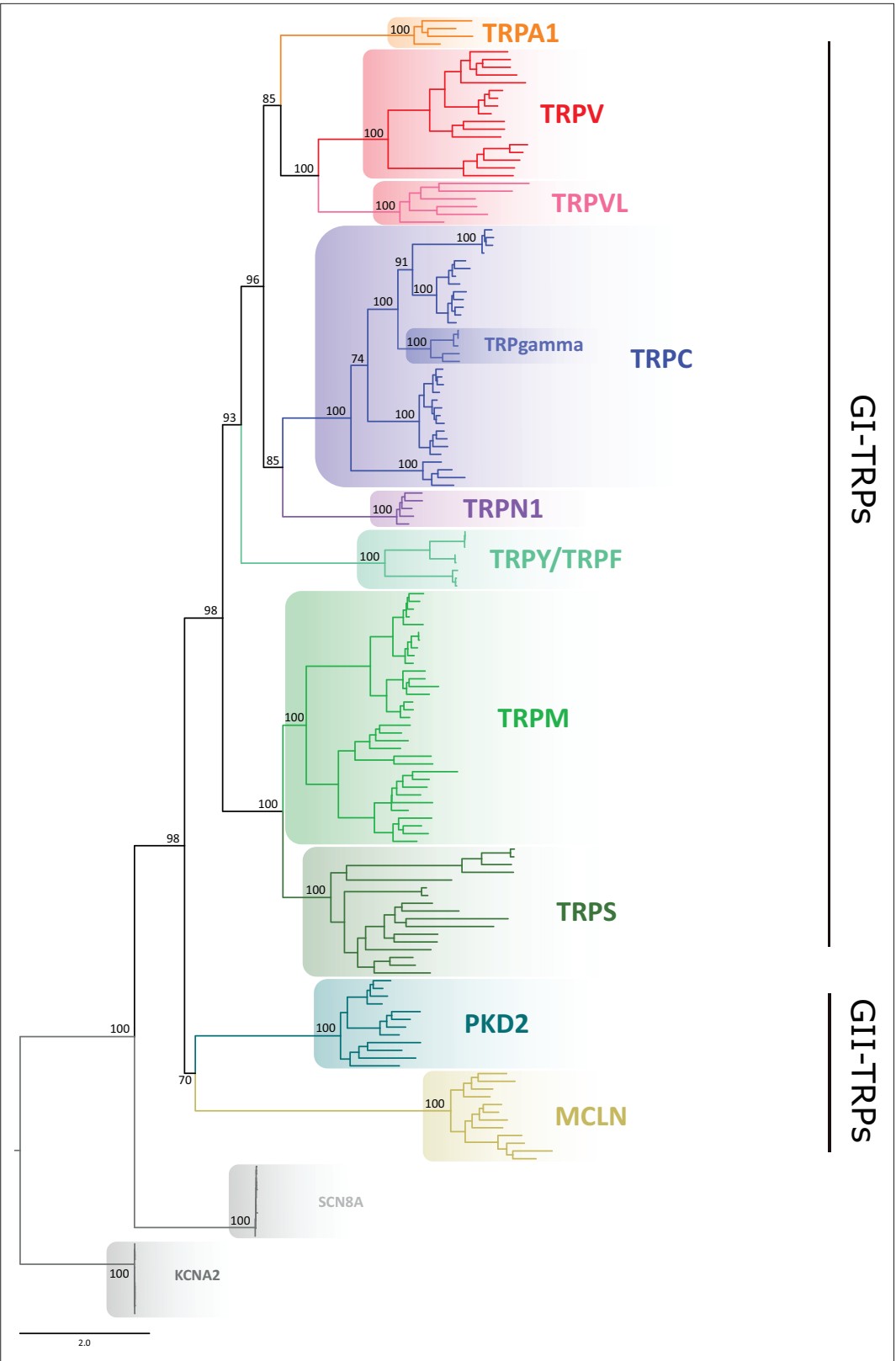

**Figure 1.** Maximum likelihood tree showing relationships among TRP channels.
The scale denotes substitutions per site and colors represent lineages. Numbers on the nodes correspond to support values from the ultrafast bootstrap routine. Potassium voltage-gated channel subfamily A member 2

*Figure 1 continued on next page*

*Figure 1 continued*

(KCNA2) and sodium voltage-gated channel alpha subunit 8 (SCN8A) sequences were included as an outgroup. TRP, transient receptor potential.

The online version of this article includes the following figure supplement(s) for figure 1:

**Figure supplement 1.** Maximum likelihood tree showing relationships among TRP channels with species indicated.

**Figure supplement 2.** Maximum likelihood tree showing relationships among TRP channels and the putative TRPs from unicellular organisms.

**Figure supplement 3.** Maximum likelihood tree showing relationships among TRP channels and the putative TRPs from unicellular organisms with all species indicated.

**Figure supplement 4.** Pipeline diagram.

pre-TM1 region, the transmembrane region, and the TRP domain helix (TDh). We enlarged the set of sequences used for phylogenetic analysis and constructed a multiple sequence alignment (MSA) using a set of bona fide TRP orthologs. The final set contains 861 sequences from vertebrates pulled from the orthologous matrix project (*Altenhoff et al., 2021*), plus 620 non-redundant TRP channel sequences gathered from the UniProt database, containing representatives of invertebrates and unicellular organisms (*Figure 1—figure supplement 4*). Amino acid frequency histograms were used to define conserved regions and structural features were mapped according to the structural data available (*Figure 2a*). In agreement with previous structural alignment studies (*Huffer et al., 2020*), we observed that TM regions are well defined and conserved (*Figure 2a*). Moreover, we observed that gaps detected in the alignment are mostly confined to linkers in between TM segments. This includes linkers, pre-TM1 to TM1, TM1–TM2, TM3–TM4, and the outer pore regions surrounding the selectivity filter. In contrast, gaps are almost non-existent at the intracellular linker between TM4 and TM5, the elbow connecting TM6 with the TDh, and the TDh itself (*Figure 2a*). The pattern of these gaps in the loops connecting transmembrane segments seems to be a good predictor for the subfamily grouping (*Figure 2—figure supplement 1*). This is the case of the longer linker between preTM1 and TM1 that is characteristic of TRPM channels, or the extended TM3–TM4 loop seen in channels from the TRPC family (*Figure 2—figure supplement 1*). Moreover, the region surrounding the selectivity filter shows diversification. While TRPV, TRPC, and TRPN channels display a larger linker between TM5 and the selectivity filter/pore helix, TRPMs show a larger linker region between the selectivity filter/pore helix and TM6 (*Figure 2—figure supplement 1*).

A thorough analysis of the distribution of amino acids identified a discrete set of highly conserved residues (identity>90%) common to the TRP superfamily (*Figure 2a and b*). About 69% of TRP channels analyzed have 12 conserved and non-contiguous amino acids we interpreted as an amino acid signature or fingerprint (**W F Φ G Φ Φ Φ N L I A W**; where Φ could be either Phe or Tyr; *Figure 2c*). We observed that 95% of the TRP channels contained at least nine of these conserved side chains (*Figure 2c*; *Table 1*). In contrast, the bacterial potassium channel KvAP displays five conserved residues in relatively similar positions while Kv1.2 exhibits only three (*Table 1*). Given the fact that the overall dynamics and mechanism of action of the S1–S4 domain in voltage-gated channels is different from that of TRP channels it is unclear for us whether the apparent conservation of TRP fingerprint residues is valid for the case of voltage-gated channels. Further work would be needed to trace the evolutionary importance of these particular residues in the context of the extended family of 6TM ion channels.

This fingerprint was used to compare the conservation between TRP channel subtypes (*Figure 2c* and *Figure 2—figure supplement 3*). While TRPVs, TRPCs, and TRPMs conserve the complete set of 12 residues (**W F Φ G Φ F Φ N L I A W**), TRPNs show 11 (**W F Y G Φ Y Φ N L** V **A W**), TRPA1 10 (**W F Y G F F F N L I** G R), and the more distantly related TRPY only 8 residues. The differences observed in TRPA1 can be mapped to the TDh. The latter is considered an important modulator of TRP gating and although different in sequence in TRPA1 channels when compared to other GI-TRPs, it is structurally equivalent (*Paulsen et al., 2015*). Thus, our results suggest that such specific specialization in TRPA originated after the divergence from the common ancestor shared with TRPVs and is unlikely to be found outside this group. Within the unicellular algae set crTRP1, which shares a common ancestor

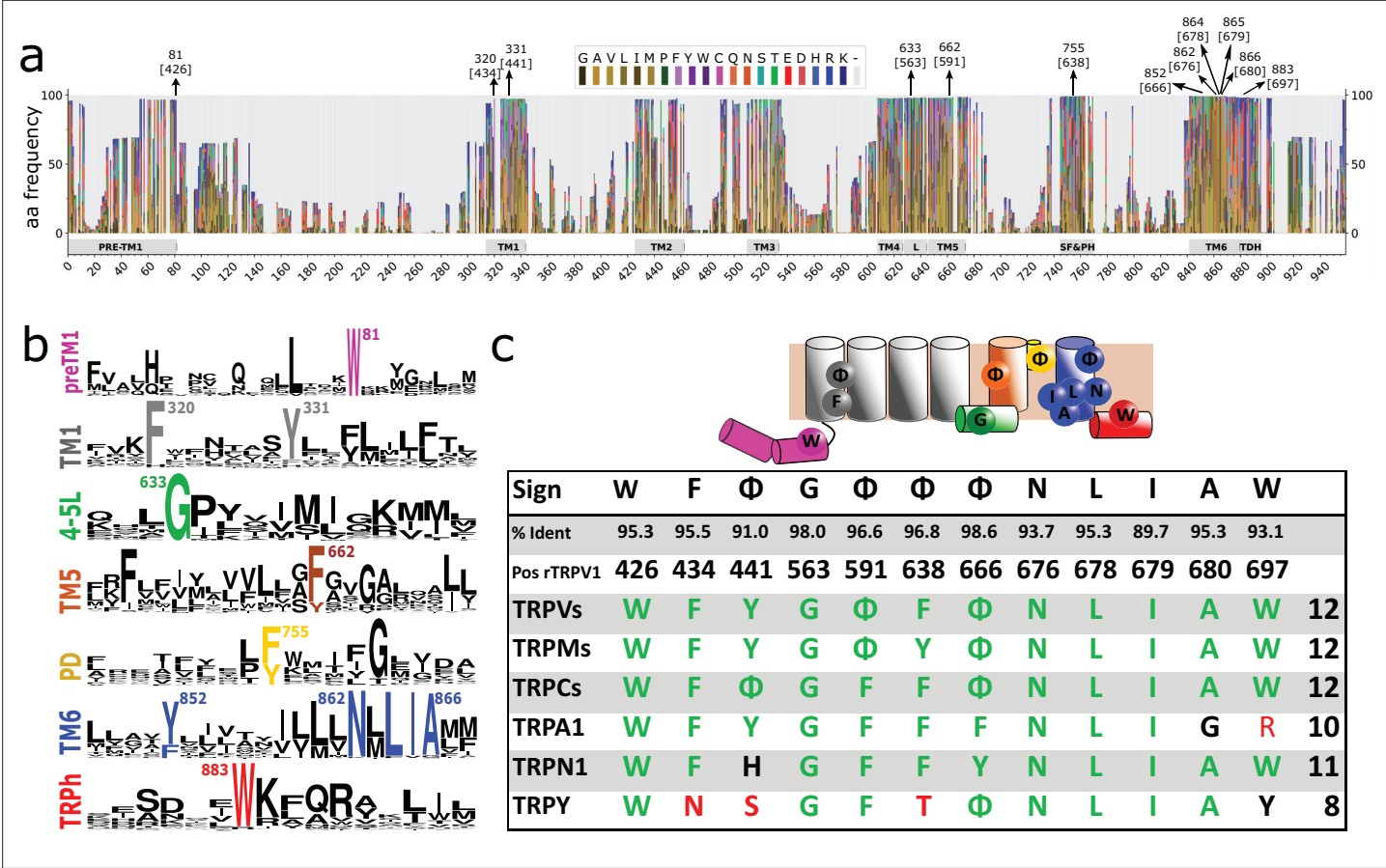

**Figure 2.** Conserved residues in GI-TRPs. (**a**) Stacked histogram showing the amino acidic probability in each position of the MAFFT alignment. Gray boxes depict the trans-membrane helices (TM1–TM6) and features such as pre-TM1, the TM4–TM5 linker (L), the Selectivity Filter and Pore Helix (SF&PH) and the TRP domain helix (TDH). Numbers over the arrows indicate the position in the alignment, and in brackets the corresponding position in the rat TRPV1 primary sequence. (**b**) Sequence logos for the TRP family, depicting highly conserved residues (>90% identity). (**c**) Upper: Cartoon of a TRP channel monomer depicting the location of conserved residues in the secondary structure. Φ denotes six carbon aromatic residues (i.e., Tyr or Phe). Bottom: Table summarizing the highly conserved positions in alignment and in the corresponding position in the rat TRPV1 primary sequence, along with the percentage of identity. Consensus residues for each subfamily are indicated. The last column corresponds to the total number of fingerprint residues for each subfamily. Green residues correspond to identities while black represents homology. Red shades denote non-conserved residues. TRP, transient receptor potential.

The online version of this article includes the following figure supplement(s) for figure 2:

**Figure supplement 1.** Stacked histograms showing the amino acid frequency in each position on the MAFFT alignment for the different TRP subgroups.

**Figure supplement 2.** Different strategies of alignment reveal the same highly conserved residues.

**Figure supplement 3.** Sequence logos for the TRP family and the analyzed subfamilies, depicting highly conserved residues (>90% identity) for the MAFFT alignment.

with the TRPM family, shows only seven fingerprint residues that increases up to nine for the case of csTRP2 that shares a common ancestor with the clade containing TRPV, TRPVL, and TRPA1 channels (*Figure 2c*; *Table 1*).

There are some highly conserved residues that were not considered in our analysis because they score just below the 90% threshold. These include a highly conserved glycine (86.3%) commonly found at the selectivity filter, a phenylalanine (85.5%) at the beginning segment of TM5, and an aspartic acid (85,2%) localized at the end of TM4–TM5 linker. The latter two residues are in close proximity to L678 (according to rTRPV1 numbering) a sidechain associated to channel response to both agonist and pH in different TRPs channels (*Boukalova et al., 2010*; *Du et al., 2009*; *Kasimova et al., 2017*; *Klausen*

**Table 1.** Table summarizing the percentage of identity of highly conserved positions in the alignment and in the corresponding positions in rTRPV1, pm TRPM8, mTRPC5, hTRPA1, and dmTRPN1.

Residues in the corresponding position of the two unicellular GI-TRPs identified (i.e., CrTRP1 and CsTRP1) are indicated. Corresponding residues in GII-TRPs and non-TRP channels are also indicated. The last column corresponds to the total number of fingerprint residues for consensus. Residues in solid black correspond to identities while italics represents homology. Red shades denote non-conserved residues.

| | W | F | Φ | G | Φ | Φ | Φ | N | L | I | A | W | |
|---|---|---|---|---|---|---|---|---|---|---|---|---|---|
| % Ident | 95.3 | 95.5 | 91.0 | 98.0 | 96.6 | 96.8 | 98.6 | 93.7 | 95.3 | 89.7 | 95.3 | 93.1 | |
| Pos Align | 81 | 320 | 331 | 633 | 662 | 755 | 852 | 862 | 864 | 865 | 866 | 883 | |
| Pos rTRPV1 | 426 | 434 | 441 | 563 | 591 | 638 | 666 | 676 | 678 | 679 | 680 | 697 | |
| Pos pmTRPM8 | 677 | 733 | 740 | 848 | 875 | 902 | 957 | 967 | 969 | 970 | 971 | 988 | |
| Pos mTRPC5 | 315 | 367 | 374 | 504 | 531 | 576 | 608 | 618 | 620 | 621 | 622 | 639 | |
| Pos hTRPA1 | 711 | 716 | 726 | 857 | 884 | 909 | 944 | 954 | 956 | 957 | 958 | 975 | |
| Pos dmTRPN1 | 1260 | 1304 | 1311 | 1427 | 1454 | 1501 | 1541 | 1551 | 1553 | 1554 | 1555 | 1572 | |
| TRPVs | W | F | Y | G | Φ | F | Φ | N | L | I | A | W | 12 |
| TRPMs | W | F | Y | G | Φ | Y | Φ | N | L | I | A | W | 12 |
| TRPCs | W | F | Φ | G | F | F | Φ | N | L | I | A | W | 12 |
| TRPA1 | W | F | Y | G | F | F | F | N | L | I | G | R | 10 |
| TRPN1 | W | F | H | G | F | F | Y | N | L | I | A | W | 11 |
| TRPY | W | N | S | G | F | T | Φ | N | L | I | A | Y | 8 |
| TRPS | W | Φ | Y | G | G | W | Y | T | L | F | A | W | 7 |
| TRPVL | W | - | N | G | Φ | F | W | N | F | I | A | A | 7 |
| Unicelular | | | | | | | | | | | | | |
| CrTRP1 | W | W | L | G | F | Q | F | N | F | I | A | F | 7 |
| CsTRP2 | W | W | Y | N | F | F | Y | N | L | I | A | F | 9 |
| TRP-GII | | | | | | | | | | | | | |
| hPKD2 | - | F | - | S | Y | F | F | N | F | L | A | - | 6 |
| mTRPML1 | F | F | H | N | Y | F | F | S | F | I | A | T | 6 |
| Non-TRP | | | | | | | | | | | | | |
| chTPCN1 I | W | Y | - | R | F | F | Y | N | L | L | A | L | 8 |
| Nav1.4 II | W | F | L | N | F | F | V | N | F | L | A | - | 7 |
| hTPCN2 II | W | F | Y | A | F | W | W | N | F | L | A | Q | 6 |
| hP2×3 | W | Y | Y | D | T | F | G | N | L | K | G | Y | 6 |
| Cav1.2 III | - | F | N | K | F | F | Y | N | F | V | G | C | 5 |
| Navab | - | - | - | R | F | F | F | N | V | V | A | - | 5 |
| KvAP | W | F | Y | G | - | - | - | - | V | V | C | W | 5 |
| Kv11.1 | E | Y | W | D | H | T | S | D | V | V | A | W | 3 |
| Kv1.2 | Y | F | G | G | - | - | - | P | L | S | S | - | 3 |
| Shaker | A | V | F | K | F | W | A | P | I | V | S | - | 2 |
| Cav1.2 III | T | T | F | S | F | W | A | P | I | V | S | - | 2 |

*et al., 2014*). These results were corroborated by a hidden Markov model (HMM) analysis using the same dataset (*Figure 2—figure supplement 2*).

Several groups have suggested an evolutionary relationship between TPCs and TRP channels (*Clapham and Garbers, 2005*; *Galione, 2011*). All three identified TPCs are thought to be asymmetric

(*Penny et al., 2016*; Kintzer and Stroud, 2018). In support of the argument of asymmetry, while one domain (D1) shows six coincidences with the TRP fingerprint, the other domain (D2) is more similar to eukaryotic voltage-gated sodium channels (Nav) with only five coincidences (*Table 1*). Interestingly, the monomeric bacterial channel NabAB (*Payandeh et al., 2011*) shares seven of these eleven signature residues, and the mammalian Nav1.4 exhibits only five of these residues in domain III, which holds the highest number of hits compared to other domains (*Table 1*). Thus, in our analysis, TPC and bacterial Nav channels exhibit the larger score of conservation at the fingerprint outside the TRP family. Overall, phylogenetic and primary sequence analyses provide strong support for a fingerprint in GI-TRP channels that is composed of 7–11 non-contiguous residues (**W F Φ G Φ Φ Φ N L I A W**).

## Sequence conservation highlights structural features

To visualize the position of these fingerprint residues, TRP channel structures from the different families were compared. We first display the frequency of amino acid coincidences and highlight the fingerprint in the context of TRPV1 (*Figure 3a and b*). By doing this structural mapping, we identified three well-defined clusters (hereafter referred to as patches) of fingerprint residues that are present in representative channels of the different families, including crTRP1 from green algae (*Figure 3c*). By extending our structural alignment to match with recently published results (*Huffer et al., 2020*), we not only confirmed that our sequence alignments are in full agreement with reported structural alignments, but we observed that the three-dimensional arrangement of the fingerprint is a robust feature among TRPs (*Figure 3—figure supplement 1*). To reconcile the different sequence data sets used in this work (*Figure 1—figure supplement 2*), rTRPV1 numbering was used throughout the manuscript to identify amino acid positions.

The first patch (P1) gathers several residues from a hotspot that has been historically linked to channel modulation. It is composed of side chains from pre-TM1 (Trp426 [95.3%]), the TM1 (Phe434 [95.5%]), the TM4–TM5 linker (Gly563 [98.0%]), and the TDh (Trp697 [93.1%]) (*Figure 3b*; *Table 1*). Initially proposed as critical for TRPV1 channel activation (*Gregorio-Teruel et al., 2014*), Glycine 563 at the linker and Tryptophane 697 at the TDh have been reported as a common theme in TRPs (*Table 2*). In this context, the TDh seems to operate as an integrator between various functional elements, receiving information from lipids (such as PIP2), the TM4–TM5 linker reporting changes occurring at the transmembrane region, and the coupling domain (CD) composed of the pre-TM1 helix and a helix-loop-helix (HLH) motif. This CD in TRPs is thought to participate in the functional association between the cytosolic and the transmembrane domain. The high conservation of Phe434 is underscored by the importance of the CD array that integrates with critical cytoplasmic features (*Garcia-Elias et al., 2015*; *Romero-Romero et al., 2017*; *Hofmann et al., 2017*; *Yang et al., 2018*; *Hilton et al., 2019*; *Yuan, 2019*; *Zubcevic et al., 2019*; *Cao, 2020*).

The second patch (P2) is located at the selectivity filter region and is composed of three phenyl group residues, that is, Phe591 [Φ 96.6%], Phe638 [Φ 96.8%], and Tyr666 [Φ 98.6%], located at both the TM6 helix and the pore helix (*Figure 3b*; *Table 1*). A glycine residue that forms part of the selectivity filter/upper constriction in TRPs is also highly conserved in all subtypes (86.3%) but slightly off to the 90% threshold (*Figure 2*). The high conservation of these three phenyl residues located within the selectivity filter and pore helix contrasts with the otherwise high variability observed in this region (i.e., turret and re-entry linker) (*Figure 2—figure supplement 1*).

A third patch (P3) is localized at the lower portion of the pore and is composed of a well-studied set of residues forming the lower gate (Asn676 [93.7%], Ile679 [89.7%], and Ala680 [95.3%]) as well as Leu678 [95.3%] that is facing the interface between TM5 and TM6 (*Palovcak et al., 2015*; *Figure 3b*; *Table 1*).

Finally, we identified a prevalent aromatic side chain (Tyr441 [76.5% Tyr+14.4% Phe]), located at the middle of the TM1 helix (*Figure 3b*; *Table 1*). Mutations in that position have been reported deleterious for TRPV1 channel function (*Boukalova et al., 2013*). This residue was not observed in close proximity to any other fingerprint residue (*Figure 3b and c*).

From the structural data, the most conserved interaction among fingerprint residues is between the Gly563 at the TM4–TM5 linker and Trp697 at the TDh. This observation is further supported by our evolutionary coupling analysis (ECA), showing that the highest score for putative interactions is precisely those established between the TM4–TM5 linker and the TDh (*Figure 3—figure supplement 2*). ECA also links the lower portion of TM2 to the end of the TDh. Structurally, such interaction

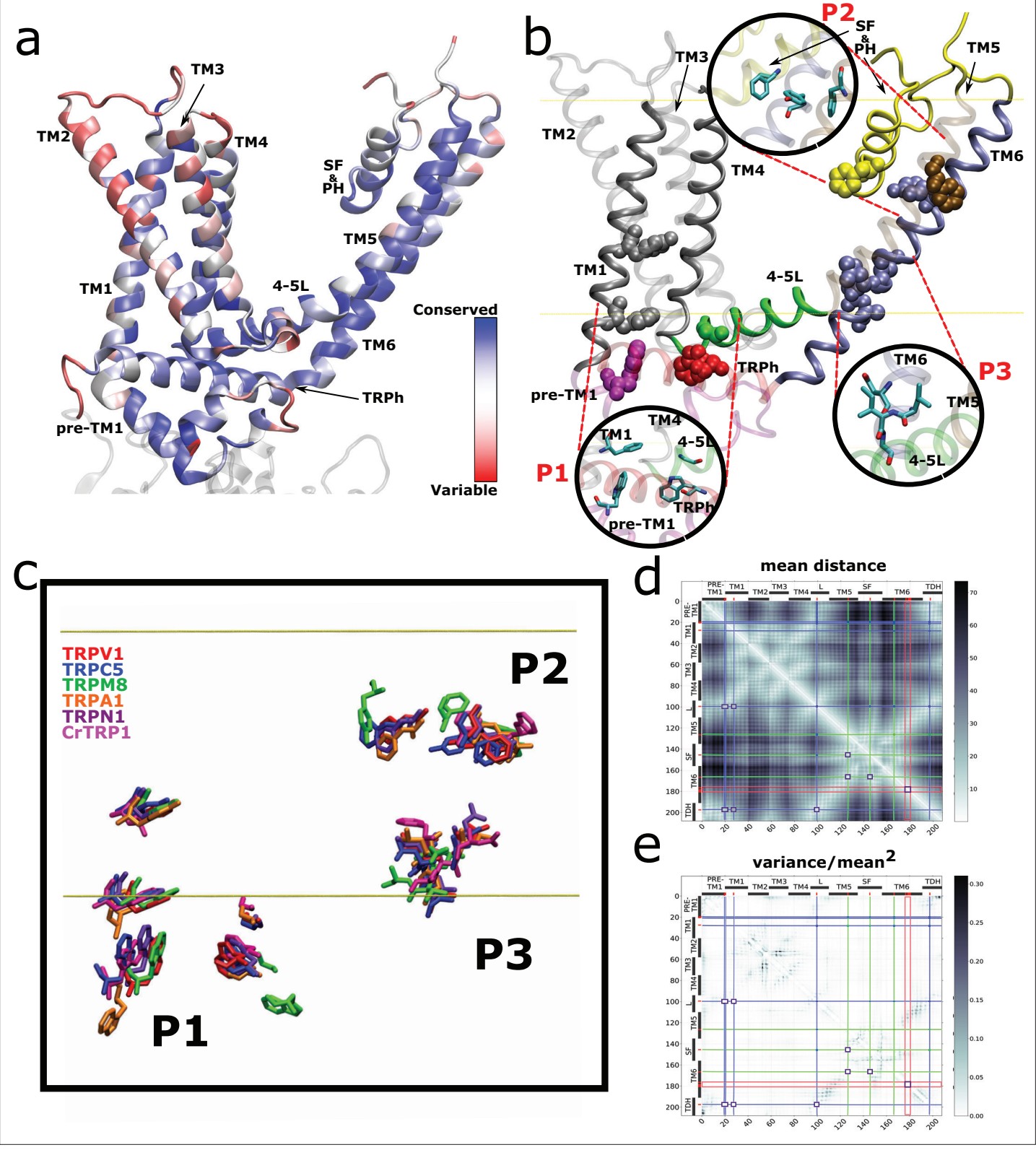

**Figure 3.** Spatial distribution of TRP channel signature residues. (**a**) Conservation rates for each position in the alignment, calculated on Consurf (see Materials and methods), mapped on rTRPV1 structure (PDB: 7LP9) (**b**) Highly conserved (>90%) residues are arranged in three well-defined patches, highlighted as insets and dubbed P1, P2, and P3. The structural data and residue numbering corresponds to rat TRPV1 (PDB: 7LP9). For clarity, only one protomer is shown. Backbone and residues follow the code color used in *Figure 2b*. 4–5L: TM4–TM5 linker; SF&PH: selectivity filter and pore helix;

*Figure 3 continued on next page*

*Figure 3 continued*

TRPh: TRP helix. (**c**) Structural alignment performed over representative channels (rTRPV1, PDB:7LP9; mTRPC5, PDB:6AEI; pmTRPM8, PDB: 6O6A; hTRPA1, PDB:3J9P; dmTRPN1, PDB:5VKQ; CrTRP1, PDB:6PW4) reveals a consistency in the position of signature residues. (**d, e**) Distogram of mean distances (**d**) and normalized variance of mean distances (**e**) between pair of residues on transmembrane segments, revealing the proximity of signature residues of same patches (brighter areas in (**d**)) and the low variability on the distances of the same pairs (brighter areas in (**e**)). Blue, green, and red lines identify the P1, P2, and P3 residues, respectively, and squares locate the intersection between these residues. TRP, transient receptor potential.

The online version of this article includes the following figure supplement(s) for figure 3:

**Figure supplement 1.** Position of signature residues in structural alignment.

**Figure supplement 2.** Coevolution analysis.

**Figure supplement 3.** Stacked histograms showing the amino acidic probability in each position on MAFFT alignment.

**Figure supplement 4.** Fingerprint residues remain at close distance.

cannot be easily explained by direct contact between residues from TM2 and TDh. Thus, it follows that the high covariance score between TM2 and the TDh could involve an additional linker molecule such as PIP2 or other lipid binding to this region (*Poblete et al., 2015*; *Yin et al., 2018*; *Yazici et al., 2021*; *Hughes et al., 2018*). Interactions between the channel and membrane lipids at the cytosol-membrane interface are emerging as a common theme in TRPs. Under this view, different parts of the CD/TDh coupling mechanism are tuned by the differences in binding of membrane lipids and/or canonical ligands (reviewed in *Zubcevic, 2020*).

## Connectivity between the signature residues

Series of independent studies had reported that mutations of residues that form part of the fingerprint—or the connecting side chains—modify or impair channel activity, underscoring their importance in maintaining proper channel activity (*Table 2*). Therefore, we analyzed the connectivity between the signature residues at the different patches using simple MSA statistics and DCA, in addition to a structural analysis. To this end, we first parsed the whole alignment by selecting only positions of high frequency (*Figure 3—figure supplement 3*) to build distance matrices for the different structures that were included in our set of sequences (138 individual structures). By averaging these individual matrices, we obtained the mean distance between the highly conserved residues (i.e., residues of high frequency that are present in all structures analyzed). Mean distance between fingerprint residues of the same patch is low, suggesting proximity as observed in the exemplary structures (*Figure 3c and d*; *Figure 3—figure supplement 4*). Moreover, the normalized variance of these pairwise distances is also low (*Figure 3e*, purple squares). This suggests that fingerprint residues on each patch remain in close proximity with each other, regardless of the variation in conformations rendered during experimental derivation. In contrast, sections of high pairwise variability are also detected. This is the case of pre-TM1, where the variance is higher for the distances with the linkers TM2–TM3 and TM4–TM5 (*Figure 3e*, darker regions). As reported abundantly in literature, the lower gate transition to the open state involves the motion of the lower portion of the TM6 (*Susankova et al., 2007*; *Salazar García et al., 2009*; *Cao et al., 2013b*). Accordingly, we can observe the large variability of distances between TM5 and TM6, an indication of the multiple conformations we included in our statistical analysis (*Figure 3e*, darker regions). Moreover, the TM1–TM4 region displays larger variability among close residues (*Figure 3e*). Notably, the amino acids belonging to the different patches are consistently close to regions of larger variability (*Figure 3e*; dark spots next to purple squares). It is important to note that our analysis surveys the landscape of possibilities observed out of a diverse collection of structural arrangements, therefore should not be over-interpreted as a direct proxy for mobility.

## A conserved AC at the transmembrane region of TRPs

Four phenyl groups (i.e., Phe/Tyr) were identified in specific positions within the transmembrane domain in more than 90% of the sequences (*Figure 4a and b*). Further inspection of structural data showed that these residues are part of a larger cluster of aromatic residues facing the center of the four-helix bundle formed by TM1–TM4, that are common to all TRP channel structures. A group of five to eight inward-facing aromatic residues belonging to the LBD domain (including at least one signature residue from P1), appear to form an AC (*Figure 4*; *Figure 4—figure supplement 1*). In contrast, voltage-gated potassium and sodium channels occupy these positions with charged amino

**Table 2.** Summary of structural-functional studies and the reported effects of site directed mutagenesis in signature residues. First column indicates the equivalent signature residue in the rTRPV1 sequence. Second column indicates the channel member studied. Third and fourth columns correspond to the type of study used to determine functional effects. MDS, molecular dynamics simulations; SDM, site-directed mutagenesis.

| TRPV1 position | Channel | Residue | Evidence source | Effect | Reference |
|---|---|---|---|---|---|
| W426 | rTRPV1 | W426A | SDM | Insensitive to Capsaicin | Zheng et al., 2018a |
| | hTRPV3 | W433 | Structure | Part of the 2-APB binding pocket | Zubcevic et al., 2019 |
| | rTRPV1 | W426A | SDM | Impaired Voltage and Capsaicin response | Zheng et al., 2018b |
| | rTRPM8 | W682A | SDM | Impaired Voltage and Menthol response | Zheng et al., 2018a |
| | hTRPA1 | W711 | Structure | Interaction site with phospholipids | Suo et al., 2020 |
| F434 | drTRPC4 | F366 | Structure | Part of cholesterol binding pocket | Vinayagam et al., 2018 |
| | faTRPM8 | F738 | Structure | Part of the Icilin and WS-12 binding pocket | Izquierdo et al., 2021 |
| F/Y441 | TRPV1 | Y441S | SDM | Nonfunctional | Boukalova et al., 2013 |
| | rTRPM8 | Y745H | SDM | Critical on Menthol Sensitivity. | Bandell et al., 2006 |
| | | Y745H | SDM | Low response to Mentol, but normal response to temperature. Critical on inhibition SKF96365-mediated of Cold- and voltage-activation, but just partially on other inhibitor | Malkia et al., 2009 |
| | | Y745H | SDM | Low response to Mentol, but normal response to temperature | Nguyen et al., 2021 |
| | hTRPC3 | Y374 | Structure | Part of the inhibitor, clemizole, binding pocket | Song et al., 2021 |
| G563 | rTRPV1 | G563S/C | SDM | Gain of Function | Boukalova et al., 2010 |
| | | G563S/A | SDM | Gain of Function, Inhibition by proton of Max current induced by capsaicin | Boukalova et al., 2013 |
| | mTRPV1 | G564S | SDM | Gain of Function | Duo et al., 2018 |
| | rTRPV3 | G573S/C | SDM | Gain of Function | Xiao et al., 2008 |
| | | G573S/C | SDM | Gain of Function, Olmsted Syndrome | Lin et al., 2012 |
| | mTRPV3 | G573S | SDM | Non responsive to Menthol, Camphor and APB and mildly responsive to temperature | Nguyen et al., 2021 |
| | rTRPV1 | G563S | SDm | Non responsive to Camphor and APB and mildly responsive to temperature | Nguyen et al., 2021 |
| | mTRPC4/5 | G503S/G504S | SDM | Gain of Function | Beck et al., 2013 |
| | hTRPC3 | G552 | Structure | Coupled W673 from TRP domain | Fan et al., 2018a |
| | hTRPC3 | G552 | Structure | Coupled W673 from TRP domain | Fan et al., 2018a |
| F/Y591 | rTRPV1 | F591 | MDS | Part of the vanilloid binding pocket | Elokely et al., 2016 |
| | | F591A | SDM | Low Capsaicin response, non response to pH and not RTX binding | Ohbuchi et al., 2016 |
| | hTRPM4 | Y944 | Structure | Forming face to face π-stack with F1027 on TM5 | Duan et al., 2018 |
| F/Y638 | rTRPV1 | F638A | SDM | Gain of Function, NMDG/Na selectivity raised | Munns et al., 2015 |
| | | F638W | SDM | Enhanced the sensitivity to the acylpolyamine toxins AG489 and AG505 | Kitaguchi and Swartz, 2005 |
| | rTRPV2 | F601 | Structure | Part of the cannabidiol binding pocket | Pumroy et al., 2019 |
| | rTRPM8 | Y908A/W | SDM | Not responsive to Cold and Menthol but responsive to Icilin | Bidaux et al., 2015 |
| | | Y908F | SDM | Totally responsive to Cold and Menthol and Icilin | Bidaux et al., 2015 |
| | zfTRPC4 | F572 | Structure | Stabilizes the pore through an hydrophobic contact with neighbor protomer | Vinayagam et al., 2018 |
| | mTRPC5 | F576A | SDM | Nonfunctional, dominant negative | Strübing et al., 2003 |
| | hTRPC5 | F576A | SDM | Differential effect on agonists: Not responsive to AM237, but responsive elgerin | Wright et al., 2020 |
| | hTRPA1 | F909A | SDM | Affect different agonists and antagonists responses | Chandrabalan et al., 2019 |

*Table 2 continued on next page*

Table 2 continued

| TRPV1 position | Channel | Residue | Evidence source | Effect | Reference |
|---|---|---|---|---|---|
| | | F909T | SDM | Abolish the A-967079-inhibition of AITC-evoked response | Paulsen et al., 2015 |
| Y/F666 | rTRPV1 | Y666A | SDM | Nonfunctional (present in membrane) | Susankova et al., 2007 |
| | mTRPV3 | Y661C | SDM | Not responsive to Temp, but responsive to agonist (2-APB and Camphor) | Grandl et al., 2008 |
| | hTRPV4 | Y702L | SDM | Not responsiveness to Temp, Agonist and Swelling | Klausen et al., 2014 |
| | hTRPM6 | Y1053C | SDM | Causes hypomagnesemia with secondary hypocalcemia, Decreased Current amplitude in heterologus expression in HEK293 | Lainez et al., 2014 |
| | hTRPM4 | F1027 | Structure | Forming face to face π-stack with Y944 on TM5 | Duan et al., 2018 |
| | hTRPA1 | F909A | SDM | Affect different agonists responses | Chandrabalan et al., 2019 |
| N676 | rTRPV1 | N676 | MDS | Gating relies on the rotatory motion of N676 | Kasimova et al., 2018 |
| | | N676A | SDM | Nonfunctional (present in membrane) | Susankova et al., 2007 |
| | | N676F | SDM | Not responsive to Temp and Agonist (Cap/RTX) and reduced response to pH | Kuzhikandathil et al., 2001 |
| | hTRPA1 | N944A | SDM | Abolished inhibition by AZ868 and A-967079, but not by HC-030031 | Klement et al., 2013 |
| L678 | rTRPV1 | L678A | SDM | Low response to Agonist (Cap) and Temp, but normal response to both at the same time | Susankova et al., 2007 |
| | | L678P | SDM | Not responsive to Temp and Agonist (Cap/RTX) and reduced response to pH | Kuzhikandathil et al., 2001 |
| | TRPV3 | L768F | SDM | Olmsted Syndrome and Erythromelalgia (gain of function) | Duchatelet et al., 2014 |
| | TRPC3 | L654 | Structure | Constriction site in the lower region of the pore | Fan et al., 2018a |
| I679 | rTRPV1 | I697 | Structure | Constriction site in the lower region of the pore | Liao et al., 2013 |
| | | I697 | Structure | Constriction site in the lower region of the pore | Cao et al., 2013a |
| | | I697 | Structure | Constriction site in the lower region of the pore | Gao et al., 2016 |
| | | I697 | Structure | Constriction site in the lower region of the pore | Chugunov et al., 2016 |
| | | I697 | Structure | Constriction site in the lower region of the pore | Susankova et al., 2007 |
| | mTRPM4 | I1036 | Structure | Constriction site in the lower region of the pore | Guo et al., 2017 |
| | hTRPM4 | I1040 | Structure | Constriction site in the lower region of the pore | Autzen et al., 2018 |
| | drTRPC4 | I617 | Structure | Constriction site in the lower region of the pore | Vinayagam et al., 2018 |
| | rTRPV4 | I715 | SDM | Hydrophobic single-residue gate. Higer resting currents | Zheng et al., 2018a |
| | mTRPC4 | I617N | SDM | Hydrophobic single-residue gate. Higer resting currents | Zheng et al., 2018b |
| | rTRPM8 | V976S | SDM | Hydrophobic single-residue gate. Higer resting currents | Zheng et al., 2018a |
| A680 | rTRPV1 | A680 | MDS | Change of Solvatation | Chugunov et al., 2016 |
| | rTRPV4 | A716S | SDM | Not responsive to agonists (4αPDD, Hypotonicity and AA), cause SMD Kozlowski type, and Metatropic Dysplasia | Krakow et al., 2009 |
| | hTRPVA1 | G955A | SDM | Slower inactivation rate. Lower rectification rates | Benedikt et al., 2009 |
| | | G958R | SDM | Inward-rectifier, constitutively active at resting potential, and impaired response to AITC | Benedikt et al., 2009 |
| W697 | rTRPV1 | W697 | Structure | It forms a hydrogen bond with the main chain carbonyl oxygen of F559 at the beginning of the S4–S5 linker | Liao et al., 2013 |
| | | W697A | SDM | Low Response to Cap/Em | Valente et al., 2008 |
| | | W697x | SDM | Low Response to Cap/Em, Affect allosteric activation | Gregorio-Teruel et al., 2014 |
| | TRPV3 | W692G | SDM | Gain of Function, Olmsted Syndrome | Lin et al., 2012 |
| | TRPV4 | W733R | SDM | Gain of Function, limited agonist response, and not inactivation to long depolarization | Teng et al., 2015 |

*Table 2 continued*

| TRPV1 position | Channel | Residue | Evidence source | Effect | Reference |
|---|---|---|---|---|---|
| | TRPC3 | W673 | Structure | *It is extensively coupled with the S4–S5 linker through interactions with G552* | **Fan et al., 2018a** |
| | TRPC4 | W674 | Structure | *Coupled with the S4–S5 linker through interactions with G553 and P546 on TM4* | **Fan et al., 2018b** |

acids forming salt bridges, short aliphatic side chains, or aromatics located at the membrane-water interface (*Figure 4a*; *Figure 4—figure supplement 1*). Although the overall three-dimensional shape of the AC varies (*Figure 4c*), it is present in all TRP subtypes, connecting three to four helices from within the LBD, suggesting they could serve as a scaffold that stitches together the whole domain.

This putative AC emerges as a common theme present from crTRP1 to TRPV1–4 (*Figure 4b*; *Figure 4—figure supplement 1*). The most dramatic case is observed in the TRPV1–4 clade, exhibiting both the highest number of aromatics (8) and the most ordered stack (*Figure 4b and c*). On the other hand, TRPA1 shows only five aromatics forming the AC without a clear stacking, as observed also in the other subfamilies (*Figure 4b*). Although the residues in TRPA1 are still in close contact, they do not form the compact stacking observed in their sister group, indicating that this might correspond to a specialization exclusive for TRPV channels (*Figure 4b and c*). In contrast to these examples, crTRP1 does not clearly form a compact-extended AC.

The presence of these conserved aromatics next to ligand-binding sites, or even forming part of them, suggests a mechanism in which the AC acts by imparting rigidity to the region and functionally linking the transmembrane helices 1–4, facilitating the translation of mechanical force from the ligand-binding sites to the CDs that connect multiple parts of the channel including the TDh, cytoplasmic CD, and pre-TM1.

Consistent with our hypothesis, we observed a large variability in the distances between residues located within the TM1–TM4 region (*Figure 4—figure supplement 2a*). However, minimal or no change in mean distance was observed among the conserved aromatics (*Figure 4—figure supplement 2a,b*). This suggests to us that their role might not be related to being part of a switch but rather stitching channel machinery in place.

The analysis of the most represented subfamilies (TRPV, TRPM, and TRPC) shows that the connectivity and association between amino acids in this region has been shaped differently (*Figure 4—figure supplement 2c-h*). Considering the role of the region in ligand binding, our sequence and structure-based analyses support the notion of progressive structural transitions enabling the functional specialization of the LBD within each different subfamily.

Finally, consistent with the notion that TPC channels are close relatives of the TRP family, the presence of a group of three aromatics facing inside the VSLD is conserved in TPC's domain 1 and absent in domain 2 that is devoid of an extended aromatic network, resembling Nav channels (analyzed on mTPC1, PDB:6C96; NaV1.4, PDB:6AGF). Unlike for TRPs, the AC in TPC channels looks less cohesive, or 'disconnected' (*Figure 4c*).

## Conserved residues at the interaction between subunits

TRP channels show a domain-swapped configuration (*Liao et al., 2013*). That is, the VSLD/LBD of one subunit appears in close contact with its pore domain (PD) through the protein backbone and, at the same time, in close contact with the PD of the neighboring subunit. The coupling between the LBD and the PD from different subunits of the tetramer is one topic that has been poorly studied in TRP channels and not well understood in VGIC (*Carvalho-de-Souza and Bezanilla, 2019*; *Shem-Ad et al., 2013*; *Neale et al., 2003*).

By inspecting the structural data, we found a conserved interaction formed by a residue at the middle of TM4 and a residue that is consistently preceding a signature residue in TM5 (Phe591 in rTRPV1; *Figure 5a*). Although the nature of the interaction varies, it is present in all surveyed structures (*Figure 5b*; *Figure 5—figure supplement 1a*). Such interaction would put in direct contact TM helices 4 and 5 from different subunits (*Figure 5—figure supplement 1*). For the case of TRPV channels, this interaction might support long-range communication between patches P1 and P2 from different

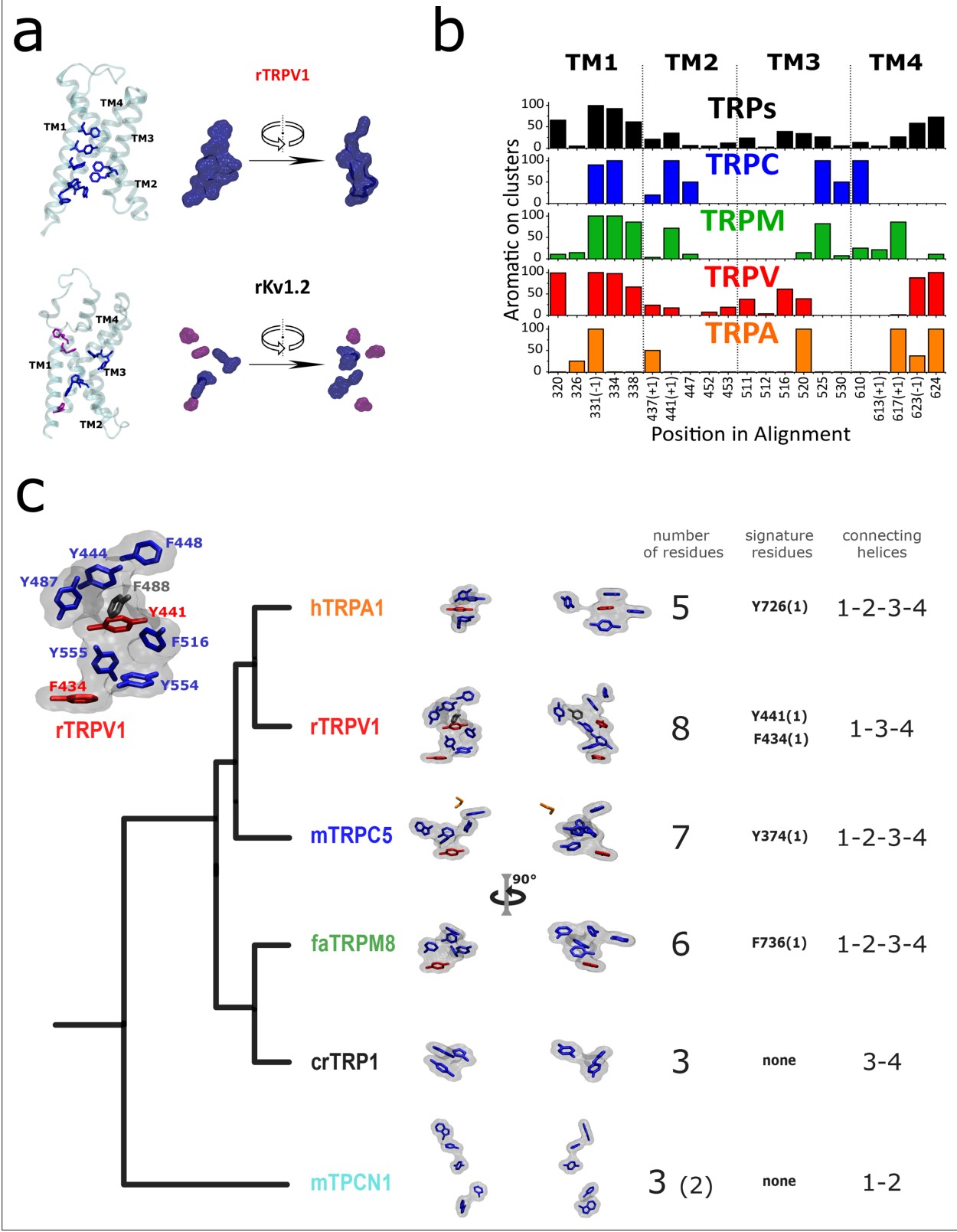

**Figure 4.** Aromatic residue distribution in LBD. (**a**) Aromatic residues facing the internal space shared by the four first transmembrane helices (core). The interacting aromatic (distance<5A) in rTRV1 (PDB:7LP9) and rKv1.2 (PDB:2R9R) are depicted in blue. In violet residues with no other aromatic at <5A. Right: surface representation of the sidechain of aromatic residues shown as licorice in the left. (**b**) Histogram of aromatic residues in the alignment, on the positions facing the core. At the bottom are depicted the positions in the alignment, and a (+1) or (−1) indicates that in one of the subfamilies the

*Figure 4 continued on next page*

*Figure 4 continued*

aromatic is immediately after or before the labeled position (shared for the rest of the subfamilies). (**c**) Comparison between AC volumes presented next to a schematic view of the topology obtained in our phylogenetic analysis. Blue: aromatic residues >50% conserved in the respective subfamily; red: aromatic residues >50% conserved in the respective subfamily and signature residue; black: not conserved residue present in the used structure; orange: not aromatic residue in the used structure, but present as an aromatic in >50% in the respective subfamily. Inset: Aromatic core in rTRPV1. The specific positions of the aromatics are indicated. Used structures: rTRPV1, PDB:7LP9; mTRPC5, PDB:6AEI; pmTRPM8, PDB: 6O6A; hTRPA1, PDB:3J9P; CrTRP1, PDB:6PW4; mTPC1, PDB:6C96. AC, aromatic core; LBD, ligand-binding domain.

The online version of this article includes the following figure supplement(s) for figure 4:

**Figure supplement 1.** Characterization of aromatic core.

**Figure supplement 2.** Distograms of mean distances and normalized variance of mean distances between pairs of residues within the TM1–TM4 region.

subunits via the aromatics running alongside the LBD. Such putative connection would be absent in subfamilies not presenting a stacked arrangement such as TRPC (*Figure 5—figure supplement 1b*). The high conservation of these residues and their relative positions within the structure suggests a common mechanism in TRPs where the selectivity filter (P2) would be functionally connected to the ligand-binding site located on a neighboring transmembrane region.

## Discussion

Here, we studied the process of diversification of TRP channels and the conservation arising from the residue frequency distribution in the transmembrane region and put them in a structural context. A highly conserved set of residues are located and grouped at strategic positions within the channel's transmembrane region, defining a fingerprint for CI-TRPs.

### Phylogenetic relationships among transient receptor potential channels

To the best of our knowledge, the complete TRP phylogeny was recently proposed by *Himmel et al., 2020*. However, their phylogenetic hypothesis is not fully resolved as it presents two trichotomies:

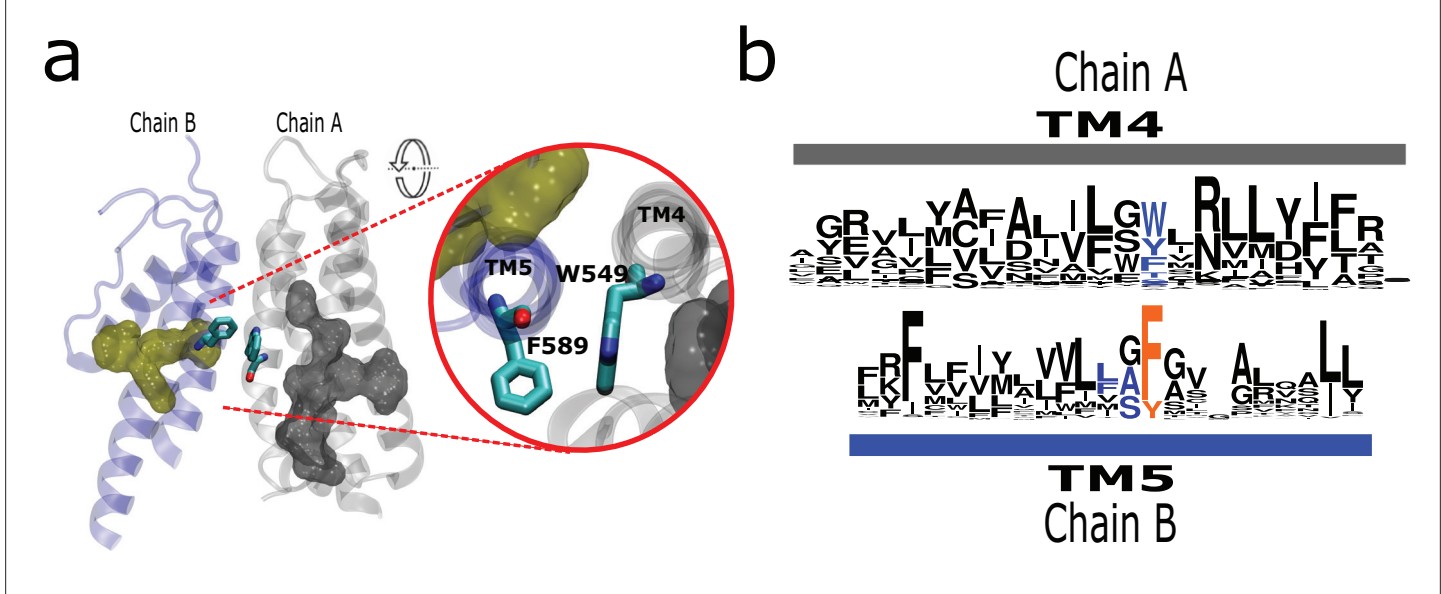

**Figure 5.** The AC connects to aromatics in P2 from neighboring subunits. (**a**) A conserved intermolecular connection between residues (licorice) in helices at opposite faces to the AC (gray surface) and P2 (yellow surface). Inset: Upper view of residues establishing the inter-subunit interaction (rTRPV1, PDB:7LP9). (**b**) Sequence logos showing the position of residues involved in the putative intermolecular interaction in blue, and the fourth signature residue in orange. AC, aromatic core.

The online version of this article includes the following figure supplement(s) for figure 5:

**Figure supplement 1.** Inter-subunit interaction is conserved in subfamilies.

the first is among TRPY/TRPF, MCLN/PKD2, and all other TRP lineages, whereas the second is among TRPA, TRPV/TRPVL, and TRPC/TRPN/TRPM/TRPS clades. Also recently, *Hsiao et al., 2021* performed an effort to define the sister group relationship among TRPs in animals, testing different methodological approaches and taxonomic samplings. However, most of their phylogenetic analyses recovered unresolved trees in addition to not including all TRP lineages (e.g., TRPY/TRPF) (*Hsiao et al., 2021*). In our case, the proposed phylogenetic hypothesis is well supported and resolves the sister group relationships among the main groups of TRP channels. Similar to other studies, we recovered the PKD2/MCLN clade sister to all other TRP lineages (*Montell, 2005*; *Venkatachalam and Montell, 2007*; *Kozma et al., 2018*); however, it is different from the proposed scenario by *Himmel et al., 2020* in which they suggest that the TRPY/TRPF clade could belong to the group that includes PKD2 and MCLN.

Most studies do not incorporate all of the TRP lineages and outgroups, making it difficult to perform direct comparisons. Regardless, our phylogenetic arrangement shows some differences with the results reported in the literature. For example, in some studies, the clade containing TRPM sequences has been recovered sister to the TRPC/TRPN1 clade (*Montell, 2005*; *Venkatachalam and Montell, 2007*; *Hsiao et al., 2021*). Other studies suggest that TRPC is sister to TRPM, and the clade containing TRPN1 sequences is the sister group of the TRPC/TRPM clade (*Arias-Darraz et al., 2015*; *Ferreira et al., 2015*; *Kozma et al., 2018*). There are also studies that show the sister group relationship between TRPV and TRPC and this clade sister to TRPM (*Clapham et al., 2001*). The TRPA1 gene lineage has been recovered sister to the TRPM/TRPC clade (*Clapham and Garbers, 2005*), to the TRPN1 clade (*Latorre et al., 2009*; *Nilius and Owsianik, 2011*; *Eriksson et al., 2018*), to the TRPV clade (*Ferreira et al., 2015*; *Peng et al., 2015*; *Kozma et al., 2018*), and to the TRPV/TRPC/TRPM/TRPN1 clade (*Montell, 2005*; *Venkatachalam and Montell, 2007*; *Hsiao et al., 2021*).

Here, we want to sound a note of caution about the use of TRP names in a taxon-specific manner because represent the common ground of this debate. The way we should 'assign names' to genes in different taxonomic groups must be based on our understanding of the duplicate history of the group of genes we are interested in (*Gabaldón, 2008*; *Altenhoff et al., 2018*). To do this, we need to perform studies including a broad and balanced taxonomic sampling and appropriate outgroups, where the reconciliation of the gene tree with the species tree plays a fundamental role (*Goodman et al., 1979*). In summary, we present a phylogenetic hypothesis for the sister group relationships of TRP channels that was inferred including all members of the gene family and outgroups. We believe that our tree topology can serve to understand the diversity and speciation of structural attributes present in the different subfamilies of TRP channels (*Figure 6*).

## A robust signature for TRP channels

A consistent architectural picture is needed to extend and generalize the multiple observations provided by structural biology. In this context, we report here a highly conserved set of residues are located at strategic positions. From the identified signature residues, those that belong to P1 and P3 have been recurrently studied in the literature. In particular, residues from P1 have been proposed critical to support the interaction between the TDh and the TM4–TM5 linker acting as an allosteric integrator (*Taberner et al., 2014*; *Sierra-Valdez et al., 2018*; *Romero-Romero et al., 2017*; *Gregorio-Teruel et al., 2014*; *Zhao et al., 2020*). On the other hand, residues from P3 have been associated with the function of the lower gate of TRP channels. In particular, to participate in stabilizing the transition between α- and π-helix types during opening (*Palovcak et al., 2015*; *Kasimova et al., 2017*; *Kasimova et al., 2018*). This observation—the critical role of amino acids in P1 and P2—holds true under the light of the structural analysis presented in here.

In contrast, the set of aromatics forming P2 are not described in the literature as such. Nevertheless, previous studies showed the importance of the residues composing this patch. Phe591 is described as forming part of the higher, wider region of the capsaicin pocket binding in TRPV1 channels (*Elokely et al., 2016*) and mutations at Tyr638 and Tyr666 have obvious effects on channel activity. Specifically, Tyr666Ala renders non-functional TRPV1 channels (*Susankova et al., 2007*), the equivalent mutant Tyr661Cys in TRPV3 is not activated by temperature, but still responds to agonists (*Grandl et al., 2008*), and the T702L equivalent mutant in TRPV4 has significantly reduced responses to agonists, temperature, and mechanical stimulation (*Klausen et al., 2014*). Moreover, mutations at equivalent positions to rTRPV1 Tyr638 in channels from different subfamilies elicit a wide range of

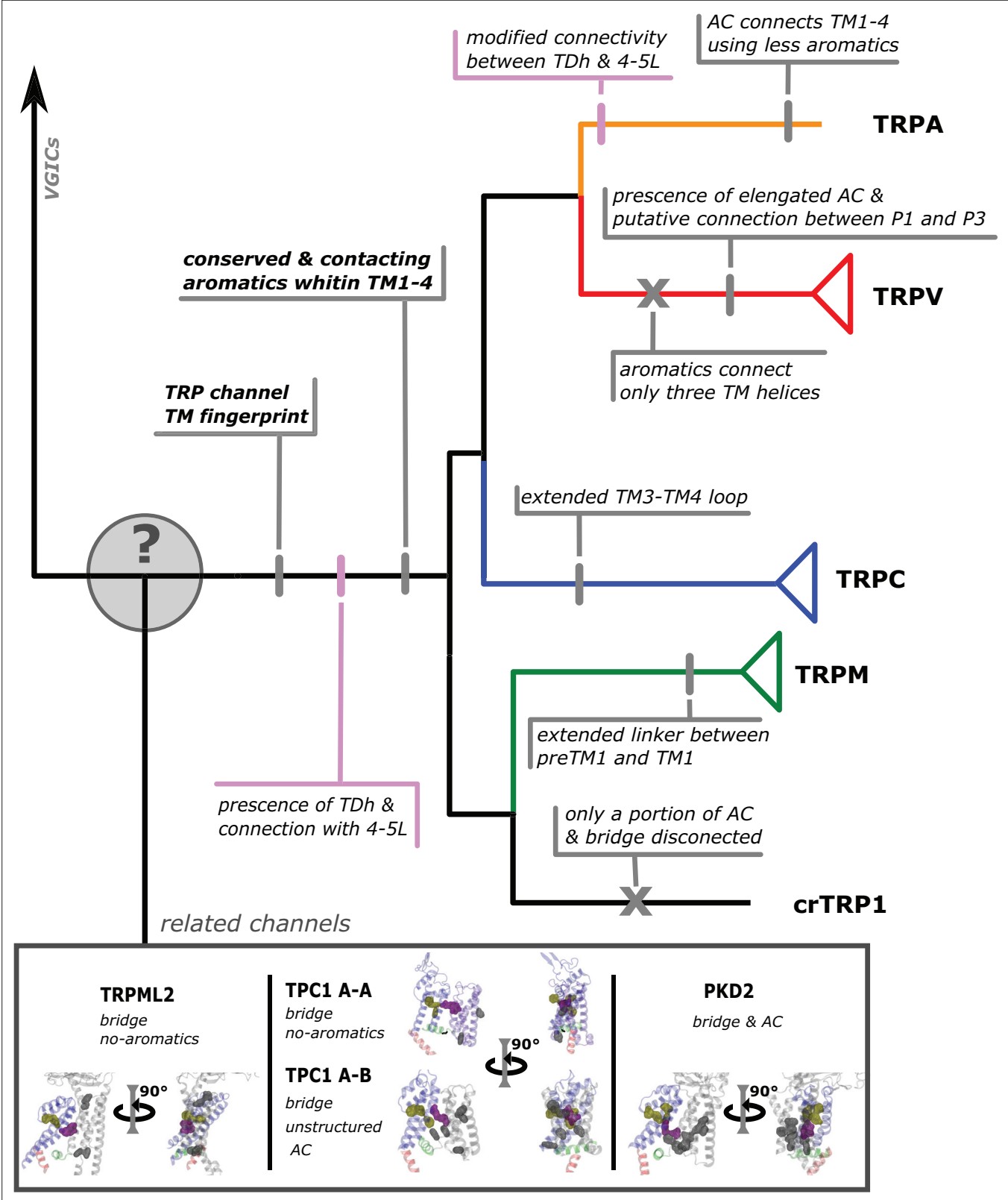

**Figure 6.** Conserved modifications observed in TRP proteins.
The different channels studied in this work are presented next to a schematic view of the topology obtained in our phylogenetic analysis. Unique TRP features are highlighted. Previous observations confirmed here are indicated in pink shades. Novel observations from the present work are indicated in gray shades. Lines represent presence while crosses represent absence or loss. TRP, transient receptor potential.

effects from gain-of-function to dominant-negative phenotypes (*Munns et al., 2015*; *Kitaguchi and Swartz, 2005*; *Bidaux et al., 2015*; *Strübing et al., 2003*; *Chandrabalan et al., 2019*; *Vinayagam et al., 2018*; *Paulsen et al., 2015*). Noteworthy, *Bidaux et al., 2015* demonstrated that mutations at this position (Tyr908 in TRPM8) to Ala or Trp abolish the response to temperature and menthol but not to icilin, yet mutation to Phe keeps the channel fully functional. We observed that the three aromatics forming P2 always need a non-conserved fourth hydrophobic residue to connect them all. By comparing apo and ligand-bound structures, P2 consistently seems to translate as if it were a near-rigid-body, moving as a whole compact motif. The function of this patch might be associated with the communication between the ligand binding domain and the state of selectivity filter in an inter-subunit fashion.

## The interaction between signature residues and the AC

Two of the residues in the P1 patch have been proposed as fundamental to the interaction between the TDh and the TM4–TM5 linker (i.e., Gly 563 and Trp 697 in TRPV1), acting as an allosteric integrator (*Taberner et al., 2014*; *Sierra-Valdez et al., 2018*; *Romero-Romero et al., 2017*; *Gregorio-Teruel et al., 2014*; *Zhao et al., 2020*; *Table 2*). A third residue in P1 corresponds to a conserved aromatic at the lower end of TM1 (i.e, Phe 441 in TRPV1) that is always connected to the other elements of the patch. Separated by two intermediate interactions, TRPM8 displays the largest distance between these two components of patch P1 (i.e., TM1 and TM4–TM5 linker/TDh). This larger distance is consistent among the TRPM family as depicted in the distance histograms. Notably, the gap observed between these two components has been associated to the binding site of menthol in TRPM8 channels (*Bandell et al., 2006*; *Malkia et al., 2009*). Moreover, the menthol analog WS-12 binds to Tyr745 (TM1), and sits close to Tyr1004 (TDh), seemingly reinforcing the connection between the LBD and the end of the TDh in TRPM8 (*Yin et al., 2019*).

There is only one signature residue that is consistently forming part of the AC (i.e., Tyr441 in TRPV1). Mutations to Ser of Tyr441 in TRPV1 generate nonfunctional channels (*Boukalova et al., 2013*). The equivalent residue in TRPM3 channels presents impairments in response to agonists when mutated from Tyr to Thr, and a Tyr to His mutation in TRPM8 channels has been shown critical to both menthol activation and the inhibitory effect of the small molecule SKF96365 (*Bandell et al., 2006*; *Malkia et al., 2009*). This coincided with similar phenotypes observed by mutating other residues belonging to the AC (*Table 3*). This is the case of Tyr444 that in TRPV1 generates nonfunctional channels (*Boukalova et al., 2013*) and the double mutation Y885T/W982R in TRPM8 that presented altered responses to agonists and temperature (*Held et al., 2018*).

When comparing the disposition of the AC in detail, it appears obvious they have different distributions in the different TRP channel families. However, certain generalizations can be drawn. The AC appears associated with P2 at the selectivity filter and disconnected from P1 in most families. Notably, in TRPVs, the AC extends down to reach the lower aromatic residues of P1 making a direct connection between the AC and the TDh/TM4–TM5 linker. Hence, a P2-AC-P1 'continuum' can be observed in the TRPV subgroup. Interestingly, from the structures obtained in the presence of agonists, we noticed that these agonists bridge the AC and P1 (e.g., WS-12 in TRPM8 and GFB-9289 in TRPC4; PDBIDs 6NR2 and 7B16, respectively; *Yin et al., 2019*; *Vinayagam et al., 2020*). The disposition of the components of P1 (i.e., preTM1; TM1; TM4–TM5 linker; TDh) and differences in the AC suggest similar but nonidentical coupling strategies among GI-TRPs. This is reinforced by the diversity of mean distance variability obtained for the different families within the TM1–TM4 region.

The presence of an AC, connecting the transmembrane helices of the ligand binding pocket and communicating critical modulatory regions that are far apart in the structure (e.g., the selectivity filter and the TDh) somewhat remind us of the case of Cys-loop receptors where an intra-membrane aromatic network contributes to the assembly and function of the receptor via interactions of both nearby and far apart residues (*Haeger et al., 2010*, 2009; *Tang and Lummis, 2018*). The AC in TRPs appears as a modular connector that has been subject to variations throughout evolution and suggests a common aspect of TRP channel mechanics that remains largely unexplored. Our line of reasoning also suggests that the observed rearrangements of the selectivity filter during activation are somewhat linked to inter-subunit interactions, likely modulated by the AC network. At the same time, it implies that certain ligands might have the ability to modulate the conformation of the selectivity filter, and by extension the extracellular linkers, without the need of an open gate conformation.

**Table 3.** Summary of mutation effects reported in the literature for residues forming part of the AC and the conserved residue at TM4 connecting TM4 with TM5.

First column indicates the equivalent signature residue in the rTRPV1 sequence. Second column indicates the channel studied. Third row corresponds to the effect of the mutation and/or proposed function.

| TRPV1 position | Channel | Mutation | Effect | Reference |
|---|---|---|---|---|
| F/Y444 | rTRPV1 | Y444S | Nonfunctional | Boukalova et al., 2013 |
| | mTRPM3 | Y885T | Impaired non-canonical current induced by pregnenolone sulfate +clotrimazol | Held et al., 2018 |
| | | | | |
| F448 | rTRPV1 | F448L | Decreased pH response but maintain all Cap responsiveness | Boukalova et al., 2013 |
| | mTRPM3 | Y888T | Similar to wt response to pregnenolone sulfate +clotrimazol | Held et al., 2018 |
| | | | | |
| Y/F554 | rTRPV1 | Y554A | Nonfunctional (Cap 10 µM, −70 to 200 mV, 48 °C) | Boukalova et al., 2010 |
| | | Y554F | Normal responsiveness | Boukalova et al., 2010 |
| | | Y554A | Not responsiveness to pH, Cap and RTX | Elokely et al., 2016 |
| | | Y554A | Increased sensitivity and affinity to 2-APB | Singh et al., 2018a |
| | rTRPV2 | Y514A | Increased sensitivity and affinity to 2-APB | Singh et al., 2018b |
| | rTRPV3 | Y564A | Increased affinity to 2-APB | Singh et al., 2018a |
| | | | | |
| Y/F555 | rTRPV1 | Y555S | Nonfunctional (Cap 10 µM, −70 to 200 mV, 48 °C) | Boukalova et al., 2010 |
| | | Y555F | Normal responsiveness | Boukalova et al., 2010 |
| | | | | |
| W549 | rTRPV1 | W549A | Not responsive to Cap (and RTX an others) and pH | Ohbuchi et al., 2016 |
| | rTRPV1 | W549A | Interaction with vanillyl moiety of RTX or Capsaicin | Gavva et al., 2004 |
| | hTRPV4 | W568A | Impaired responsiveness to heat and agonists (4α-PDD and BAA); but responsive to swelling and endogen lipids | Vriens et al., 2007 |
| | mTRPM3 | W982R | Abolished non-canonical current induced by pregnenolone sulfate +clotrimazol | Held et al., 2018 |
| | mTRPM3 | W982F | Similar to wt response to pregnenolone sulfate +clotrimazol | Held et al., 2018 |
| | hTRPA1 | Y840F | Reduced potency of ligand (GNE551) | Liu et al., 2021 |
| | hTRPA1 | Y840W/H/L/A | Completely abolished potence of ligand (GNE551) | Liu et al., 2021 |
| | hTRPA1 | Y840A | Impaired response to AITC and almost abolished to β-Eudesmol | Ohara et al., 2015 |

## Temperature-dependent gating from a phylogenetic and structural perspective

TRP channel's ability to respond to temperature is a property that has been reported in 11 TRP channels TRPV1–4, TRPA1, TRPM2, TRPM3, TRPM4, TRPM5, TRPM8, and TRPC5 (*Caterina et al., 1997*; *Saito and Shingai, 2006*; *Saito et al., 2011*; *Ferreira et al., 2015*; *Saito and Tominaga, 2015*; *Castillo et al., 2018*). Despite the strong conservation of the phenotype across GI-TRPs, we have failed to find a conserved domain common to all temperature-sensitive TRP channels within the region analyzed in the present study.

Considering the phylogenetic relationships among TRPs, the phenotype distribution suggests a scenario involving multiple gain and/or losses of the ability to respond to temperature. The temperature-sensitive crTRP1 (*Arias-Darraz et al., 2015*; *McGoldrick et al., 2019*), from the unicellular algae *C. reinhardtii*, shares a common ancestor with the TRPM subfamily, that includes at least five out of eight members displaying different degrees of temperature sensitivity. The feature is apparently lost in TRPCs, with the notable exception of TRPC5 (*Zimmermann et al., 2011*). Would be hard at this time to elaborate whether the origin of temperature sensitivity in TRPC5 channels represents

a gain or loss of a functional trait. In contrast, temperature-dependent gating is strongly conserved in the TRPV group, with the exception of calcium-selective TRPV5 and TRPV6. The phenotype was apparently preserved in TRPA1 during a process of specialization that involved multiple sequence changes in the so called 'allosteric nexus' region (*Paulsen et al., 2015*), while preserving the core structural features defining GI-TRP channels presented in here.

Recent structural work (*Nadezhdin et al., 2021b*; *Kwon et al., 2021*), anisotropic thermal diffusion calculations (*Diaz-Franulic et al., 2016*), and a combination of patch clamp fluorometry, mutagenesis, and molecular modeling (*Yang et al., 2018*) suggest that the temperature-dependent transition may initiate close to the intracellular water-lipid boundary and propagate through a conformational wave. Interestingly, this collective molecular motion is enabled by a network of interactions involving fingerprint residues Trp426 and Phe434, which connect to the transmembrane helices via the AC and the TM4–TM5 linker/TDh interface. The conserved residues and interactions highlighted in the present work suggest new and exciting mutagenesis experiments that could potentially dissect the temperature-driven conformational wave and thus shed light on the microscopic mechanism of heat activation.

## Materials and methods
### Amino acid sequences, alignments, and phylogenetic analyses

To advance our understanding of the sister group relationships among TRPs, we retrieve amino acid sequences from the National Center for Biotechnology Information (NCBI) (PMID: 29140470) corresponding to TRPVs, TRPVL, TRPA1, TRPC, TRPgamma, TRPN1, TRPY/TRPF, TRPM, TRPS, PKD2, and MCLNs lineages. In most cases, we included representative species of vertebrates (TRPA1, TRPVs, TRPCs, TRPN1, TRPMs, PKD2s, and MCLNs). However, in the case in which the TRP lineages are not present in vertebrates, we included representative species of the groups in which the TRP channel is present. Thus, TRPVL included cnidarians and annelids, TRPgamma included insects, arachnids, and merostoms, TRPS included nematodes, chordates, arachnids, chilopods, priapulids, cephalopods, bivalves, and tardigrades, TRPY/TRPF had several species of fungi. Accession numbers and details about the taxonomic sampling are available in *Supplementary file 1*. Amino acid sequences were aligned using MAFFT v.7 (*Katoh et al., 2019*) allowing the program to choose the alignment strategy (FFT-NS-i). We used the proposed model tool of IQ-Tree v.1.6.12 (*Minh et al., 2020*) to select the best-fitting model of amino acid substitution (JTT+F+I+G4). We used the maximum likelihood method to obtain the best tree using the program IQ-Tree v1.6.12 (*Minh et al., 2020*). We performed five tree searches in which the initial gene tree was provided by ourselves, which was previously estimated using IQ-Tree v.1.6.12 (*Minh et al., 2020*). We also carried out five additional analyses in which we performed more exhaustive tree searches by modifying the strength of the perturbation (-pers) from 0.5 (default value) to 0.9 and the number of unsuccessful iterations to stop (-nstop) from 100 (default value) to 500. The tree with the highest likelihood score was chosen. We assessed support for the nodes using the ultrafast bootstrap routine as implemented in IQ-Tree v1.6.12 (PMID: 29077904). Potassium voltage-gated channel subfamily A member 2 (KCNA2) and sodium voltage-gated channel alpha subunit 8 (SCN8A) amino acid sequences from mammals were included as an outgroup (*Supplementary file 1*). Our next step was to retrieve TRP amino acid sequences from *C. reinhardtii*, *Volvox carteri*, *C. subellipsoidea*, *Micromonas pusilla*, *Dictyostelium discoideum*, *Dictyostelium purpureum*, *Leishmania infantum*, *Leishmaniamajor*, *Leishmania mexicana*, *Paramecium tetraurelia*, and *Trypnosoma cruzi*. To do so, the transmembrane regions corresponding to TRPA1, TRPC1, TRPC3–7, TRPM1–8, TRPML1–3, TRPP1–3, and TRPV1–6 from human (*Homo sapiens*), TRPC2 from the house mouse (*Mus musculus*), NompC from the fruit fly (*Drosophila melanogaster*), and TRPY1 from the brewing yeast (*Saccharomyces cerevisiae*) were used as queries in blastp searches (*Altschul et al., 1990*) against the proteomes of above-mentioned species. Putative channels were selected based on the frequency of hits to the query sequences relative to human, fruit fly, and yeast. This was followed by reciprocal blastp searches (E-value<1e), and a final inspection for the presence of the TRP domain. To investigate the phylogenetic position of these candidate TRP channels, we aligned them with the sequences previously sampled which includes all main groups of TRP channels. Amino acid sequences were aligned using MAFFT v.7 (*Kintzer and Stroud, 2018*) allowing the program to choose the alignment strategy (FFT-NS-i). We used the proposed model tool of IQ-Tree v.1.6.12 (*Minh et al.,*

*2020*) to select the best-fitting model of amino acid substitution (LG+G4). We used the maximum likelihood method to obtain the best tree using the program IQ-Tree v1.6.12 (*Minh et al., 2020*). We performed five tree searches in which the initial gene tree was provided by ourselves, which was previously estimated using IQ-Tree v.1.6.12 (*Minh et al., 2020*). We also carried out five additional analyses in which we performed more exhaustive tree searches by modifying the strength of the perturbation (-pers) from 0.5 (default value) to 0.9 and the number of unsuccessful iterations to stop (-nstop) from 100 (default value) to 500. The tree with the highest likelihood score was chosen. We assessed support for the nodes using the ultrafast bootstrap routine as implemented in IQ-Tree v1.6.12 (PMID: 29077904). Potassium voltage-gated channel subfamily A member 2 (KCNA2) and SCN8A amino acid sequences from mammals were included as an outgroup (*Supplementary file 1*).

## MSA database

We retrieved 969 protein sequences corresponding to the subgroups TRPV, TRPA1, TRPM, TRPN, and TRPC in representative species of all main lineages of amniotes from the Orthologous Matrix project (OMA) (*Altenhoff et al., 2021*). About 646 extra sequences (including those of unicellular TRP) from the Uniprot protein database were added to the pool of sequences rescued from the OMA server and then aligned using MAFFT (FFTNS1 strategy). The region corresponding to the transmembrane was identified, and the rest of the sequence was removed, leaving the section from the last portion of the pre-TM1 region to the last residue of the TRP helix (residues 396–718, using rTRPV1 as reference). A second round of sequence alignment in MAFFT (L-INS-I strategy) was performed and manually refined to minimize gaps. The resulting MSA—Primary MSA—database contains 1481 monophyletic sequences, 861 selected from OMA database and 620 from Uniprot, including sequences annotated as TRP or TRP-like channels and several uncharacterized protein sequences (*Figure 1—figure supplement 2*). In order to facilitate the visualization by minimizing even more the gaps, TRPS and TRPVL sequences were taken out for the generation of figures but remained for the statistical processing.

## Hidden Markov models

The fingerprint TRP residues in *Table 1* were identified from sequence and structural alignments. To further corroborate this conservation analysis, we defined a profile HMM and analyzed the emission probabilities at each position. For each matching position of the HMM, emission probabilities describe a generalized Bernoulli distribution: conserved positions are characterized by a distribution peaked around the invariant amino acid. We use Shannon entropy to quantify the 'peakedness' of the distribution. We create separate HMMs from the MSAs of the major TRP protein subfamilies, and also a cumulative HMM that includes all of these MSAs together. To define the matching positions, we selected only the MSA columns with less than 50% of gaps. To train the HMM, we used hmmbuild from the HMMER suite. The HMM predictions for the fingerprint residues in each protein subfamily match those in *Table 1*, which confirms the identification of the fingerprint residues in that table. The 12 signature residues are among the low entropy positions in the MSA, indicating that the HMM has determined they are indeed well-conserved (*Figure 2—figure supplement 2b*).

## Structural alignment

Structural alignment in *Figure 3C* was performed in the inbuilt extension of VMD, aligning the first and sixth transmembrane helices, following the numbering provided by the MSA.

## Coevolution analysis

Coevolution scores were calculated using asymmetric pseudo-likelihood maximization direct coupling analysis algorithm (aplmDCA) (*Ekeberg et al., 2013*). This algorithm finds the approximate parameters of the maximum entropy probabilistic model consistent with selected MSA statistics (univariate and bivariate frequency distributions). Default parameters were used for field and coupling regularization and sequence reweighting (lamba_h=0.01, lambda_j=0.01, and theta=0.1).

## Distance matrices
### Data

PFAM provides deletion-free (insert-only) MSAs whose sequences they have aligned to PDBs on a residue-by-residue basis. In order to align sequences in this way, the resulting MSA has a high number

of gaps. Our structural analysis includes 140 TRP sequence-structure pairs across multiple TRP subfamilies, which were already pre-indexed and mapped by PFAM, and then subsequently verified by us. The original MSA was 958 long, and after feature selection and verification, 208 positions remained from 91 sequences.

### Feature selection

To narrow down which positions to use, we removed positions with high gap frequency, which are potentially not shared across the TRP subfamilies. Accordingly, we considered only MSA positions present in more than 96% of the sequences. This means that the tolerable gap frequency across the entire alignment for a position was <4%, ensuring that the positions selected in this way were present across all subfamilies. We created a frequency histogram of the full MSA, which showed which positions of the MSA have a high frequency of gaps. We also used knowledge-based feature selection to isolate positions that fall within key regions of functional importance: pre-TM1, TM1, TM2, TM3, TM4, Linker, TM5, Pore, TM6, and TDH.

### Distograms

From the feature selected sequence-structure maps, pairwise distance matrices were computed from the PDB structures using Cβ-Cβ residue distances (or Cα for glycine). Distograms for mean, variance, and normalized variance were computed across the distance matrices and visualized. Not only do they show which pairwise positions vary by distance, but they also show which pairwise distances are conserved, across the subfamilies.

### Interpretation

The mean distograms are the easiest to interpret because they simply give an overall picture of which residue pairs are nearby, or distant, to others across the subfamilies. Mean distograms showed expected α-helical structure and reasonable separation between the different helical elements. For example, whereas TM1–TM4 are observed to be close to each other as a group, they appear to be further away from TM5 to TM6, which are close to each other as a group. This is consistent with the canonical structure of 6TM channel proteins. On the other hand, variance distograms give an overall picture of variation in residue-residue distances. This information is useful to characterize specific sequence patterns within a large collection of structural states. Given the wide range of distances, thermal fluctuations and other noise sources are expected to produce the largest variances for residue-pairs that are separated by large distances. To remove this bias, variances are normalized, that is, they are divided by the square of the average distance (from the distance matrix). By this procedure we effectively highlight variance of residues that are at short distances.

## Identification of residues at the AC

A python script was used to identify the aromatic residues facing the internal cavity formed by the TM1 to TM4 helixes in channel structures extracted from PDB files. For this, the distances between all alpha carbons in opposing helices (TM1–TM3 and TM2–TM4) were calculated, choosing the nearest residue pairs. The same procedure was followed for gamma/alpha pairs. Those residues with (Cα-Cα) distance>(Cα-CY)distance were recorded in a list. After, we calculated distances between all atoms in pairs of aromatic residues in this list, and kept the minimal distances between pairs. Then, a third code grouped in clusters all the pairs with a distance below a threshold of 5 Å. Scripts and files used are available in https://github.com/brauchilab/ProteinCoreCluster, copy archived at swh:1:rev:a934ed-29d9e77d34a19a47158f8819b373de5842; *Cabezas-Bratesco, 2022*.

## Figure preparation

To visualize the identities and gap patterns on the MSA, images were exported using Jalview 2.11.1.3 (*Waterhouse et al., 2009*). The sequence logos show the distribution of amino acid residues at each position in the regions of interest, and were generated using WebLogo version 3 (*Crooks et al., 2004*). The structural figures were generated using VMD 1.9.2 (*Humphrey et al., 1996*), using the files next PDB files: 7LP9 for rTRPV1; 6O6A for paTRPM8; 6AEI for hTRPC5; 3J9P for hTRPA; 5VKQ for dmTRPN1; 6PW4 for crTRP1; 6C96 for mTPCN1; and 2R9R for rKv1.2. Direct interactions between

residues were identified using a distance threshold of 4 Å, except for the pi-pi interactions, where a 5 Å threshold was used (*Piovesan et al., 2016*).

## Acknowledgements

The authors acknowledge Dr. Gonzalo Riadi from Universidad de Talca for providing computer support. JCO acknowledges the Integrative Biology Group members, Universidad Austral de Chile, for their constant support, scientific enthusiasm, and creative feedback. This work was supported by Fondo Nacional de Desarrollo Científico y Tecnológico from Chile (FONDECYT 3140233) to CKC, (FONDECYT 1191868) to SB, (FONDECYT 1210471) to JCO, and ANID-Millennium Science Initiative Program #NCN19_168 (SB and JCO). The Millennium Nucleus of Ion Channel-Associated Diseases (MiNICAD) is a Millennium Nucleus of the Iniciativa Milenio, National Agency of Research and Development (ANID, Chile). This work was funded in part by the National Institutes of Health (R01GM093290, S10OD020095, and R01GM131048; VC), and the National Science Foundation through Grant no. IOS-1934848 (VC). This research includes calculations carried out on HPC resources supported in part by the National Science Foundation through major research instrumentation grant number 1625061 as well as by the US Army Research Laboratory under contract number W911NF-16-2-0189 (VC).

## Additional information

### Funding

| Funder | Grant reference number | Author |
|---|---|---|
| Fondo Nacional de Desarrollo Científico y Tecnológico | FONDECYT 3140233 | Charlotte K Colenso |
| Fondo Nacional de Desarrollo Científico y Tecnológico | FONDECYT 1191868 | Sebastian E Brauchi |
| Fondo Nacional de Desarrollo Científico y Tecnológico | FONDECYT 1210471 | Juan C Opazo |
| Agencia Nacional de Investigación y Desarrollo | Millennium Science Initiative Program #NCN19_168 | Juan C Opazo Sebastian E Brauchi |
| National Institutes of Health | R01GM093290 | Vincenzo Carnevale |
| National Science Foundation | IOS-1934848 | Vincenzo Carnevale |
| National Science Foundation | 1625061 | Vincenzo Carnevale |
| U.S. Army | Research Laboratory W911NF-16-2-0189 | Vincenzo Carnevale |
| National Institutes of Health | S10OD020095 | Vincenzo Carnevale |
| National Institutes of Health | R01GM131048 | Vincenzo Carnevale |

The funders had no role in study design, data collection and interpretation, or the decision to submit the work for publication.

### Author contributions

Deny Cabezas-Bratesco, Conceptualization, Formal analysis, Investigation, Methodology, Writing – original draft; Francisco A Mcgee, Formal analysis, Methodology, Software, Writing – review and editing; Charlotte K Colenso, Conceptualization, Data curation, Formal analysis, Investigation, Methodology, Writing – review and editing; Kattina Zavala, Data curation, Formal analysis, Investigation,

Methodology; Daniele Granata, Formal analysis; Vincenzo Carnevale, Data curation, Formal analysis, Investigation, Methodology, Writing – review and editing; Juan C Opazo, Data curation, Formal analysis, Investigation, Methodology, Writing – original draft, Writing – review and editing; Sebastian E Brauchi, Conceptualization, Data curation, Methodology, Project administration, Writing – original draft, Writing – review and editing

Author ORCIDs
Vincenzo Carnevale ⓘ http://orcid.org/0000-0002-1918-8280
Juan C Opazo ⓘ http://orcid.org/0000-0001-7938-4083
Sebastian E Brauchi ⓘ http://orcid.org/0000-0002-8494-9912

Decision letter and Author response
Decision letter https://doi.org/10.7554/eLife.73645.sa1
Author response https://doi.org/10.7554/eLife.73645.sa2

---

## Additional files

Supplementary files
• MDAR checklist
• Supplementary file 1. Accession numbers of the TRPs genes used in this study.

Data availability
Sequences, MSA and phylogenetic reconstruction data analyzed during this study are are available in Dryad database: https://doi.org/10.5061/dryad.k6djh9w75. Code for analysis of Aromatic Core is available in Github database: https://github.com/brauchilab/ProteinCoreCluster, (copy archived at swh:1:rev:a934ed29d9e77d34a19a47158f8819b373de5842).

The following dataset was generated:

| Author(s) | Year | Dataset title | Dataset URL | Database and Identifier |
|---|---|---|---|---|
| Cabezas-Bratesco D, McGee FA, Colenso C, Zavala K, Granata D, Carnevale V, Opazo JC, Brauchi SE | 2022 | Sequence conservation and structural features that are common within TRP channels | https://dx.doi.org/10.5061/dryad.k6djh9w75 | Dryad Digital Repository, 10.5061/dryad.k6djh9w75 |

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
