## [Editor Report]

This study by Deny Cabezas-Bratesco and collaborators draws from multiple bioinformatics approaches, as well as from published structural and functional data, to uncover a set of highly conserved amino acid sequence features in group I TRP ion channels. These identified features provide insight into the evolution and mechanisms of function of this diverse and important family of ion channel proteins.

---

## [Decision Letter]

**Decision letter after peer review:**

Thank you for submitting your article "Sequence conservation and structural features that are common within TRP channels" for consideration by *eLife*. Your article has been reviewed by 3 peer reviewers, including Andrés Jara-Oseguera as Reviewing Editor and Reviewer #1, and the evaluation has been overseen by Richard Aldrich as the Senior Editor.

The reviewers have discussed their reviews with one another, and the Reviewing Editor has drafted this to help you prepare a revised submission. All three reviewers agreed that the findings in the manuscript are relevant and interesting for a wide audience, and that data has been carefully analyzed. However, they also raised a series of concerns that would need to be addressed. Below is a list of essential revisions agreed upon by all reviewers, together with the individual reviews with additional helpful suggestions.

Essential revisions:

1) The authors should provide additional discussion regarding TRP subfamilies that have been identified more recently in unicellular organisms and invertebrates (TRPVL, TRPS, TRPF/TRPY, TRPL/TRP-γ), and consider them as separate subfamilies in their discussion, or otherwise justify why they chose to group them together with one of the vertebrate subfamilies. The authors should confirm whether sequences from these families were already included in their analysis, and include them if they were not considered initially.

2) The authors should include a cladogram or a similar graphic that allows readers to assess sequences from which organisms were included in the analysis, and whether different groups of organisms are similarly represented within the analyzed sample.

3) The authors should describe how the boundaries of the individual TM domains were determined, and how many structures were used to find these boundaries.

4) Figure 2A is very hard to read and confers limited information. The authors should include a new supplementary figure containing the entire MSA with sufficient resolution to appreciate the amino acid letters in the sequences. I suggest substituting Figure 2A with a different graphic that better highlights the differences in the gaps between subfamilies – perhaps showing histograms of gaps per amino acid position for each family computed from the sequences could be informative.

5) Data presentation in Figure 2 – Supplement 1 should be improved, as it is very hard to read. I suggest the authors show the MSA per subfamily but keeping the lengths of the gaps that arise from the MSA containing all families – this could provide a clearer depiction of the relations between gaps and subfamilies. It is also hard to appreciate the features that group the TRPL and TRP-γ sequences with TRPC channels – I suggest the authors include the sequences of TRPL and TRP-γ in the MSA for the other subfamilies, so that the higher similarity with the TRPC family can be better appreciated.

6) Lines 129-130: "usefulness of these patterns as predictors by looking at *Drosophila* channels TRPL and TRPgamma that were included in the dataset pulled from Uniprot." Why these particular channels? Also, if they were used in the dataset that generated the results is it fair to use them to test the results? It seems tautological. Is this test really needed?

7) HMM techniques are very well-known – the authors must have considered them along the way: why did the authors chose not to use them in the end? The task of finding conserved residues starting from one or several multiple sequence alignments could benefit from using HMM-based methods. These methods associate to each MSA a "profile", a probabilistic version of a consensus sequence. Different profiles can be compared via HMM-HMM alignments, which gives a pretty good description of the most relevant difference among the starting MSAs. It would be interesting to compute an HMM profile for each TRP family and look at these differences. This will for sure contain and hopefully extend your considerations about the gap "bar code".

8) How different are the trees produced by the maximum likelihood algorithm? I agree with the authors' choice of keeping the tree with the greatest likelihood (provided the MSA is fixed), but at the same time it would be interesting to look at how different the results are, in order to have a rough estimate of the reliability of the method.

9) The authors do not include branch lengths in their trees. This is a fundamental aspect for algorithms only capable of generating binary trees, and could be useful in reinforcing the points the authors make on the clades they find.

10) The functional significance of the aromatic core is not convincingly determined beyond that it might provide stability to the protein. The bulk of the functional studies actually suggest that mutating these residues only rarely abolishes coupling of ligand binding to gating, as proposed. The authors should more clearly discuss the potential relevance that the hydrophobic core might have on channel function – it would strengthen the manuscript if the authors provided a set of predictions or experimentally testable hypothesis for the functional relevance of this core. The authors should perform a thorough search of the literature for perturbations introduced in that region to test whether the published results are consistent with their hypothesis, and discuss their findings accordingly. The finding of a conserved interaction between residues on the TM4 and TM5 (W549 and F589) is significant, but there is no functional data for F589 or structural comparison analysis of the two sites in the apo and bound states – the authors should provide additional information from the literature if available.

11) Early on it is noted that the pre-S1 is functionally important, but findings for this region were not reported. The authors should include a brief note on those regions that are functionally important but yielded no highly conserved residues.

12) The authors should add labels for the transmembrane helices in Figure 3 and Figure 3 – Supplement 2. Without the labels, it is very hard to follow the discussion about the role of lipids in bridging regions with co-evolving residues. Have lipids been observed in structures specifically at those positions? The authors should discuss this more clearly.

13) The authors should provide stronger support for the conservation of signature residues in non-TRP channels. This could be done by including a sequence alignment for those channels or structural data such as that in Figure 3 – Supplement 1.

14) Structure-based sequence alignments are publicly available from the work by Huffer et al., (*eLife*, 2020) that is cited by the authors. The authors should analyze or at least discuss whether the signature residues they identify in their MSA also align in the structure-based alignments. This would provide stronger support for the structural conservation of the signature residues.

15) Figure 3 – Supplement 2 is inadequate to show changes in connectivity in the signature residues. The authors should introduce this topic with more nuance regarding the uncertainties in interpreting structural differences in single residues when the functional states that are represented by the structures are not known, and when the conformational differences that can be observed between a pair of aligned structures strongly depend on which regions of the protein were chosen for the alignment – the entire tetramer vs a single subunit vs just the pore- or the S1-S4-domains. The authors should analyze the structural data in a more systematic way, including data from more than one representative from each subfamily.

16) Figure 1: denote what UC stands for in the caption.

17) Line 172: please specify what TPCN stands for.

18) Figure 4: please clarify if the violet asterisk is between subfamilies? Rather than within. What does “AC” stand for?

19) Line 288: define VGIC.

20) In Figure 6 is it unclear whether the position of the markers for each evolutionary event has any significance, and if it does, what support is there for it. Perhaps reducing the size of the phylogenetic tree and including similar structural schemes for each group I TRP channel subfamily, as done for the outliers, could make the figure more informative.

*Reviewer #1 (Recommendations for the authors):*

1) The authors should include a cladogram or a similar graphic that allows readers to assess the diversity and balance between organisms from which sequences were analyzed.

2) Figure 2A is very hard to read and confers limited information. The authors should include a new supplementary figure ontainning the entire MSA with sufficient resolution to appreciate the amino acid letters in the sequences. I suggest substituting Figure 2A with a different graphic that better highlights the differences in the gaps between subfamilies – perhaps showing histograms of gaps per amino acid position for each family computed from the sequences could be informative.

3) Figure 2 – Supplement 1 is also very hard to read. I suggest the authors show the MSA per subfamily but keeping the lengths of the gaps that arise from the MSA containing all families – this would provide a better illustration of the relations between gaps and subfamilies. It is also hard to appreciate the features that group the TRPL and TRP-γ sequences with TRPC channels – I suggest the authors include the sequences of TRPL and TRP-γ in the MSA for the other subfamilies, so that the contrast can be better appreciated. This aspect of the manuscript could be strengthened if the authors provided data showing that the patterns of sequence gaps allow a clustering algorithm to correctly segregate channel ‘barcodes’ into their correct subfamilies.

4) The authors should add labels for the transmembrane helices in Figure 3 and Figure 3 – Supplement 2. Without the labels, it is very hard to follow the discussion about the role of lipids in bridging regions with co-evolving residues. Have lipids been observed in structures specifically at those positions? The authors should discuss this more clearly.

5) The authors should provide stronger support for the conservation of signature residues in non-TRP channels. This could be done by including a sequence alignment for those channels or structural data such as that in Figure 3 – Supplement 1.

6) Structure-based sequence alignments are publicly available from the work by Huffer et al., (*eLife*, 2020) that is cited by the authors. The authors should analyze or at least discuss whether the signature residues they identify in their MSA also align in the structure-based alignments. This would provide stronger support for the structural conservation of the signature residues.

7) Figure 3 – Supplement 2 is inadequate to show changes in connectivity in the signature residues. The authors should introduce this topic with more nuance regarding the uncertainties in interpreting structural differences, and discuss more at length their specific choices of structures for the analysis, and the implications of excluding certain structures and channel subtypes.

8) In Figure 6 is it unclear whether the position of the markers for each evolutionary event has any significance, and if it does, what support is there for it. Perhaps reducing the size of the phylogenetic tree and including similar structural schemes for each group I TRP channel subfamily, as done for the outliers, could make the figure more informative.

9) "a sidechain associated to channel response to both agonist and pH in different TRPs channels" – there are different mechanisms by which pH modulates TRP channels, and protons function as agonists for some of them. I suggest making this statement more specific.

10) "certain ligands might have the ability to modulate the conformation of the selectivity filter and by extension the extracellular linkers- without the need of an open gate conformation." – the significance of this statement is unclear.

*Reviewer #2 (Recommendations for the authors):*

Suggesting that you have determined the "phylogenetic position of unicellular TRP channels" yet excluding TRPVL and TRPS channels to me makes this analysis less compelling. TRPF/TRPY were included but this was not mentioned anywhere except the methods. Given this, it cannot be determined whether these novel channels are associated with known TRP subfamilies.

The functional significance of the aromatic core is not convincingly determined beyond that it might provide stability to the protein. The bulk of the functional studies actually suggest that mutating these residues only rarely abolishes coupling of ligand binding to gating, as proposed. Comparisons of the apo and ligand-bound state suggest it does not move upon channel opening. I also am not convinced that a similar set of non-rotating residues does not exist in other channels just because they are not at these exact sites.

Can you compare the locations of the TM4 and TM5 residues in the apo and ligand bound structures? This finding is compelling and could be easily checked with the data already included.

It was not described how the boundaries of the individual TM domains were determined, how many structures were used to find these boundaries.

Early on it is noted that the pre-S1 is functionally important, but findings for this region were not reported.

To make this more of interest to a broad audience, more detail about the medical and biological significance of TRP channels might be appropriate in the introduction.

*Reviewer #3 (Recommendations for the authors):*

I thank the authors for their very meticulous work for organizing the existent sources and pieces of knowledge regarding TRP proteins.

The original numerical contribution is also interesting. The bioinformatics techniques used are very standard, but they are used carefully. I have a few questions:

– The task of finding conserved residues starting from one or several multiple sequence alignments could benefit from using HMM-based methods. These methods associate to each MSA a "profile", a probabilistic version of a consensus sequence. Different profiles can be compared via HMM-HMM alignments, which gives a pretty good description of the most relevant difference among the starting MSAs. It would be interesting to compute an HMM profile for each TRP family and look at these differences. This will for sure contain and hopefully extend your considerations about the gap "bar code". HMM techniques are very well-known, thus I guess the authors must have considered them along the way: why did the authors chose not to use them in the end?

– How different are the trees produced by the maximum likelihood algorithm? I agree with the authors' choice of keeping the tree with the greatest likelihood (provided the MSA is fixed), but at the same time it would be interesting to look at how different the results are, in order to have a rough estimate of the reliability of the method.

– Also, the authors do not include branch lengths in their trees. This is a fundamental aspect for algorithms only capable of generating binary trees, and could be useful in reinforcing the points the authors make on the clades they find.

– As the authors point out, several PDB structures of TRP exist. This is a very precious resource: have the authors considered any online resource for their structural comparison?

– The authors perform a coevolution analysis, but the results are not mentioned in the Results section. Could they expand on this aspect?

---

## [Author Response]

Essential revisions:1) The authors should provide additional discussion regarding TRP subfamilies that have been identified more recently in unicellular organisms and invertebrates (TRPVL, TRPS, TRPF/TRPY, TRPL/TRP-γ), and consider them as separate subfamilies in their discussion, or otherwise justify why they chose to group them together with one of the vertebrate subfamilies. The authors should confirm whether sequences from these families were already included in their analysis, and include them if they were not considered initially.

We agree with this concern. To solve this problem, we performed new phylogenetic analyses. This time we considered a broad and balanced taxonomic sampling and included appropriate outgroups. Therefore, our phylogenetic hypothesis now reconciles better the gene tree with the species tree (as we recommend at the Discussion section lines 394 to 396). The tree is now smaller in number of sequences but better in terms of likelihood topology.

In the first analysis (Figure 1), we included all reported lineages of the TRP gene family; we think this result is relevant as today there is no clear picture of how they are related. In the second (Figure 1 —figure supplement 1), we included our unicellular sequences.

2) The authors should include a cladogram or a similar graphic that allows readers to assess sequences from which organisms were included in the analysis, and whether different groups of organisms are similarly represented within the analyzed sample.

We agree with this comment. To address this concern, in addition to the main tree presented in figure 1 (without labels for clarity), we now include a supplementary figure containing the names of the species considered in our sampling and the statistical support.

3) The authors should describe how the boundaries of the individual TM domains were determined, and how many structures were used to find these boundaries.

This is a reasonable critique raised by the reviewers. In the original version, TM segments were determined according to the boundaries established by TRPV1’s structural data and ex post confirmation by both, i) the coincidence with TM segments in few handpicked sequences having structural data that were contained in the alignment and ii) further analysis of hydrophobicity extracted from the alignment. By following this method, we observed a large consistency in the estimation of TM boundaries. In the current version we used formal structural alignments. As described in the edited method section, these were generated from a curated residue-by-residue mapping from PFAM of Uniprot sequences to PDB structures (about 100). This corresponds to the “feature selection” process described in methods and depicted in our pipeline scheme (Figure 1 —figure supplement 2).

4) Figure 2A is very hard to read and confers limited information. The authors should include a new supplementary figure containing the entire MSA with sufficient resolution to appreciate the amino acid letters in the sequences. I suggest substituting Figure 2A with a different graphic that better highlights the differences in the gaps between subfamilies – perhaps showing histograms of gaps per amino acid position for each family computed from the sequences could be informative.

After inspection we concur with the reviewers. Accordingly, we explored a better way to represent the differences among sub families. As the alignment is large, efforts to depict individual amino acids did not improve the readability of the figure. For that reason and following the recommendation this new version display histograms for PanTRPs (Figure 2a) and for the individual subfamilies in the context of the phylogenetic topology (Figure 2 —figure supplement 1). In our opinion, these greatly improve the readability of the data.

5) Data presentation in Figure 2 – Supplement 1 should be improved, as it is very hard to read. I suggest the authors show the MSA per subfamily but keeping the lengths of the gaps that arise from the MSA containing all families – this could provide a clearer depiction of the relations between gaps and subfamilies. It is also hard to appreciate the features that group the TRPL and TRP-γ sequences with TRPC channels – I suggest the authors include the sequences of TRPL and TRP-γ in the MSA for the other subfamilies, so that the higher similarity with the TRPC family can be better appreciated.

This supplemental figure was replaced. Following the recommendation this new version display histograms for the individual subfamilies in the context of the phylogenetic topology (Figure 2 —figure supplement 1). In the new figure, global differences and similarities in terms of insertions and gaps are evident.

6) Lines 129-130: "usefulness of these patterns as predictors by looking at Drosophila channels TRPL and TRPgamma that were included in the dataset pulled from Uniprot." Why these particular channels? Also, if they were used in the dataset that generated the results is it fair to use them to test the results? It seems tautological. Is this test really needed?

We eliminated this section. As mentioned by the reviewers it was not useful and ultimately confusing to the reader. We now included carefully selected orthologs from unicellular organisms and invertebrates to surmount the phylogenetic issue and in the absence of structural data helping us with mapping features, we refrained to discuss this further in the present version.

7) HMM techniques are very well-known – the authors must have considered them along the way: why did the authors chose not to use them in the end? The task of finding conserved residues starting from one or several multiple sequence alignments could benefit from using HMM-based methods. These methods associate to each MSA a "profile", a probabilistic version of a consensus sequence. Different profiles can be compared via HMM-HMM alignments, which gives a pretty good description of the most relevant difference among the starting MSAs. It would be interesting to compute an HMM profile for each TRP family and look at these differences. This will for sure contain and hopefully extend your considerations about the gap "bar code".

We agree with the reviewers, the bar code was eliminated from this version. Following the reviewer’s suggestion, we trained an HMM and analyzed the model’s parameters. The results of this investigation fully support the initial fingerprint obtained from the MSA empirical frequencies. As a bonus, we highlighted a residue at the preTM1 region that we would not have previously considered using our hard 90% threshold. This data is available in figure 2 —figure supplement 2.

8) How different are the trees produced by the maximum likelihood algorithm? I agree with the authors' choice of keeping the tree with the greatest likelihood (provided the MSA is fixed), but at the same time it would be interesting to look at how different the results are, in order to have a rough estimate of the reliability of the method.

We appreciate this comment. Regarding how different the trees are produced by the maximum likelihood algorithm; we are very satisfied because after running the analysis 10 times our results seem robust. In all ten phylogenetic analyses performed, the evolutionary relationships among the main TRP lineages were the same; further, the likelihood values varied in a small amount, between -232,213.354 to -232,212.137.

9) The authors do not include branch lengths in their trees. This is a fundamental aspect for algorithms only capable of generating binary trees, and could be useful in reinforcing the points the authors make on the clades they find.

This problem is solved; the new trees are all depicted as phylograms.

10) The functional significance of the aromatic core is not convincingly determined beyond that it might provide stability to the protein. The bulk of the functional studies actually suggest that mutating these residues only rarely abolishes coupling of ligand binding to gating, as proposed. The authors should more clearly discuss the potential relevance that the hydrophobic core might have on channel function – it would strengthen the manuscript if the authors provided a set of predictions or experimentally testable hypothesis for the functional relevance of this core. The authors should perform a thorough search of the literature for perturbations introduced in that region to test whether the published results are consistent with their hypothesis, and discuss their findings accordingly. The finding of a conserved interaction between residues on the TM4 and TM5 (W549 and F589) is significant, but there is no functional data for F589 or structural comparison analysis of the two sites in the apo and bound states – the authors should provide additional information from the literature if available.

The functional effects resulting from mutagenesis of these residues is now provided in table 3. It mentions functional impairments associated to positions Y444, F448, F554, F555, and W549 in TRPV1 and other TRPs. Certainly, a more extensive experimental validation is needed, however that is beyond the scope of our present work.

Moreover, at the discussion we included a new section (Lines 489 to 524) in which we tried to articulate a mechanism, proposing a role for all the elements mentioned in the context of temperature activation that in agreement with the current knowledge.

Lines 495 to 501:

“Under the light of recent structural work (Nadezhdin et al., 2021; Kwon et al., 2021) and anisotropic thermal diffusion calculations (Diaz-Franulic et al., 2016), it would not be unreasonable to support the notion that temperature-dependent transition initiates close to the intracellular water-lipid boundary (e.g., in the region around fingerprint residues Trp426 and Phe434), propagating throughout the AC altering both the network of interactions within the connected transmembranes and the TM4-TM5 linker/TDh interface.”

Lines 510 to 514:

“Under our working hypothesis, temperature would induce motion at the CD that propagates to the lower gate. This would be followed by the transmembrane helices forming the LBD via reshaping intra-subunit interactions supported by the AC. In turn, the rearrangement propagates via inter-subunit interactions reaching the pore and turret from a neighbor subunit.”

11) Early on it is noted that the pre-S1 is functionally important, but findings for this region were not reported. The authors should include a brief note on those regions that are functionally important but yielded no highly conserved residues.

In an effort to satisfy reviewer’s curiosity, we re-sampled all the sequences contained in the initially proposed MSA, extending the sequences towards the N-terminal so the preTM1 is included in all sequences. This can be observed in figure 2a and figure 2 -supplement figure 1. By doing this and by comparing our different strategies (i.e., frequency and HHM) we identified an additional residue (W426; now part of *patch 1*) to the fingerprint that we reported in our first version of the work.

12) The authors should add labels for the transmembrane helices in Figure 3 and Figure 3 – Supplement 2. Without the labels, it is very hard to follow the discussion about the role of lipids in bridging regions with co-evolving residues. Have lipids been observed in structures specifically at those positions? The authors should discuss this more clearly.

Lipids have been observed, the references supporting our statement are now in the text. Please see lines 255 to 263:

“ECA also links the lower portion of TM2 to the end of the TDh. Structurally, such interaction cannot be easily explained by direct contact between residues from TM2 and TDh. Thus, it follows that the high covariance score between TM2 and the TDh could involve an additional linker molecule such as PIP2 or other lipid binding to this region (Poblete et al., 2014; Yin et al., 2017; Yazici et al., 2021; Hughes et al., 2018). Interactions between the channel and membrane lipids at the cytosol-membrane interface are emerging as a common theme in TRPs. Under this view, different parts of the CD/TDh coupling mechanism are tuned by the differences in binding of membrane lipids and/or canonical ligands (reviewed in Zubcevic, 2020).”

Additionally, figure 3 – supplement figure 2 has been edited to indicate the TM helices and additional important features.

13) The authors should provide stronger support for the conservation of signature residues in non-TRP channels. This could be done by including a sequence alignment for those channels or structural data such as that in Figure 3 – Supplement 1.

We agree with this comment. Thus, we added a corresponding structural alignment as indicated in methods. The result is informed in the new table 1.

14) Structure-based sequence alignments are publicly available from the work by Huffer et al., (eLife, 2020) that is cited by the authors. The authors should analyze or at least discuss whether the signature residues they identify in their MSA also align in the structure-based alignments. This would provide stronger support for the structural conservation of the signature residues.

We repeated the exact alignment done in Huffer et al., 2020. All the signature residues identified by us are present. This result is indicated in figure 3 figure supplement 2 and commented in lines 215 to 219:

“By extending our structural alignment to match with recently published results (Huffer et al., 2020), we not only confirmed that our sequence alignments are in full agreement with reported structural alignments, but we observed that the three-dimensional arrangement of the fingerprint is a robust feature among TRPs (Figure 3 —figure supplement 1).”

15) Figure 3 – Supplement 2 is inadequate to show changes in connectivity in the signature residues. The authors should introduce this topic with more nuance regarding the uncertainties in interpreting structural differences in single residues when the functional states that are represented by the structures are not known, and when the conformational differences that can be observed between a pair of aligned structures strongly depend on which regions of the protein were chosen for the alignment – the entire tetramer vs a single subunit vs just the pore- or the S1-S4-domains. The authors should analyze the structural data in a more systematic way, including data from more than one representative from each subfamily.

We strongly agree with this critique. It was the weakest point of our initial approach. This new version of the manuscript contains a thorough structural analysis performed on distance matrices calculated for 138 individual structures. For each pair of amino acids, we analyzed the distance histogram (distogram) and reported mean and variance as summary statistics. This is showed in figure 3 panels d and e and figure 4 —figure supplement 2. The results reinforce and extend what we initially observed for selected cases.

16) Figure 1: denote what UC stands for in the caption.

It originally denoted Unicellular groups. However, the new strategy for phylogenetic reconstruction made this unnecessary. Most of the unicellular TRP channels are closer to GII-TRPs in our new phylogeny (supporting previous results from my laboratory). The only two that segregate with GI-TRPs are now mentioned by name and NCBI code.

17) Line 172: please specify what TPCN stands for.

It is defined now at the introduction in line 28.

“TRPs have been related to all voltage-gated cation channels (VGCC), two-pore channels (TPC or TPCN),…”

18) Figure 4: please clarify if the violet asterisk is between subfamilies? Rather than within. What does "AC" stand for?

After reading the critiques we realized that the original figure 4 was not easy to follow. It was reformatted for better understanding. As consequence, the violet asterisk is not present anymore.

19) Line 288: define VGIC.

It is defined now at the introduction in line 44.

“Structural data revealed that TRP channels share the general architecture of voltage-gated ion channels (VGIC)”

20) In Figure 6 is it unclear whether the position of the markers for each evolutionary event has any significance, and if it does, what support is there for it. Perhaps reducing the size of the phylogenetic tree and including similar structural schemes for each group I TRP channel subfamily, as done for the outliers, could make the figure more informative.

Figure 6 was reformatted to indicate two groups of observations. In pink shades, a group of observations that have been reported in the literature. In grey shades, observations coming from this present study. By putting the structural observations (i.e., line equals presence, cross equals absence) in the context of the tree topology (in Figure 1) we aimed to deliver a better picture of speciation in terms of structural features.

We decided not to include the structures and features of TRP channels (as in the structural depiction of characteristics present in related channels) because the figure rendered too crowded and the details were better depicted in new figures 3, 4, and 5.